# Folding of a bacterial integral outer membrane protein is initiated in the periplasm

Rakesh Sikdar[1], Janine H. Peterson[1], D. Eric Anderson[2] & Harris D. Bernstein[1]

The Bam complex promotes the insertion of β-barrel proteins into the bacterial outer membrane, but it is unclear whether it threads β-strands into the lipid bilayer in a stepwise fashion or catalyzes the insertion of pre-folded substrates. Here, to distinguish between these two possibilities, we analyze the biogenesis of UpaG, a trimeric autotransporter adhesin (TAA). TAAs consist of three identical subunits that together form a single β-barrel domain and an extracellular coiled-coil ("passenger") domain. Using site-specific photocrosslinking to obtain spatial and temporal insights into UpaG assembly, we show that UpaG β-barrel segments fold into a trimeric structure in the periplasm that persists until the termination of passenger-domain translocation. In addition to obtaining evidence that at least some β-barrel proteins begin to fold before they interact with the Bam complex, we identify several discrete steps in the assembly of a poorly characterized class of virulence factors.

[1] Genetics and Biochemistry Branch, National Institute of Diabetes and Digestive and Kidney Diseases, National Institutes of Health, Bethesda, MD 20892, USA. [2] Advanced Mass Spectrometry Facility, National Institute of Diabetes and Digestive and Kidney Diseases, National Institutes of Health, Bethesda, MD 20892, USA. Correspondence and requests for materials should be addressed to H.D.B. (email: harris_bernstein@nih.gov)

Almost all integral membrane proteins can be divided into one of two structural classes. Proteins that reside in most of the eukaryotic cell membranes or the bacterial inner membrane (IM) typically contain one or more hydrophobic α-helical membrane-spanning segments. These proteins are targeted to the endoplasmic reticulum or IM cotranslationally, and the hydrophobic membrane-spanning segments likely partition into the lipid bilayer through a lateral gate in the Sec complex in a stepwise fashion[1, 2]. In contrast, most proteins that reside in the outer membrane (OM) of Gram-negative bacteria reach their destination post-translationally and contain a unique membrane-spanning segment known as a β-barrel. This type of membrane-spanning segment is an amphipathic β-sheet that forms a closed cylindrical structure. Because β-barrels expose a hydrophobic surface that enables them to integrate into the OM only after they fold, they are presumably assembled by a distinct mechanism. The details of the assembly process, however, remain poorly understood.

Although early studies showed that periplasmic chaperones including Skp and SurA have important roles in the biogenesis of bacterial OM proteins (OMPs)[3, 4], the discovery that a hetero-oligomer called the Bam complex is required for the membrane integration of β-barrels was a major breakthrough[5, 6]. In *E. coli*, the Bam complex consists of an OMP (BamA) and four lipoproteins (BamB-E). Only BamA and BamD are conserved in Gram-negative bacteria and are essential for viability[5–9]. BamA consists of a β-barrel domain and five periplasmic POTRA (polypeptide transport associated) domains that appear to form a

flexible hinge and mediate the binding of the lipoprotein subunits[10–13]. Available evidence suggests that the POTRA domains also have a key role in substrate recognition[14].

Although the structure of the entire Bam complex was recently solved, it is still unclear that how this machine catalyzes the membrane integration of β-barrel proteins. In one model, β-strands are threaded through the pore of the BamA β-barrel in a largely unfolded conformation and then inserted (possibly in a stepwise fashion) into the lipid bilayer through a putative lateral gate that was revealed by the BamA crystal structure[15]. Indeed, the observation that the introduction of disulfide bonds into the β-barrel that lock it in a closed conformation is deleterious strongly suggests that the opening of the gate is critical for BamA function[16, 17]. In an alternative model, the Bam complex facilitates the membrane integration of folded or partially folded β-barrels by perturbing the lipid bilayer. This model is supported by evidence that BamA lowers the kinetic barrier for OMP insertion imposed by lipid head groups[18]. On the basis of a structural analysis, it has also been proposed that the BamA POTRA domains promote β-barrel formation by a β-augmentation mechanism[10]. Interestingly, the crystal structure of the Bam holocomplex shows that the POTRA domains and the four lipoproteins form a ring below the BamA β-barrel[19–21]. Although molecular dynamic simulations suggest that the ring structure induces movement of substrates by a rotational motion, they do not clarify whether the substrates pass through the pore of the BamA β-barrel or directly into a distorted lipid bilayer.

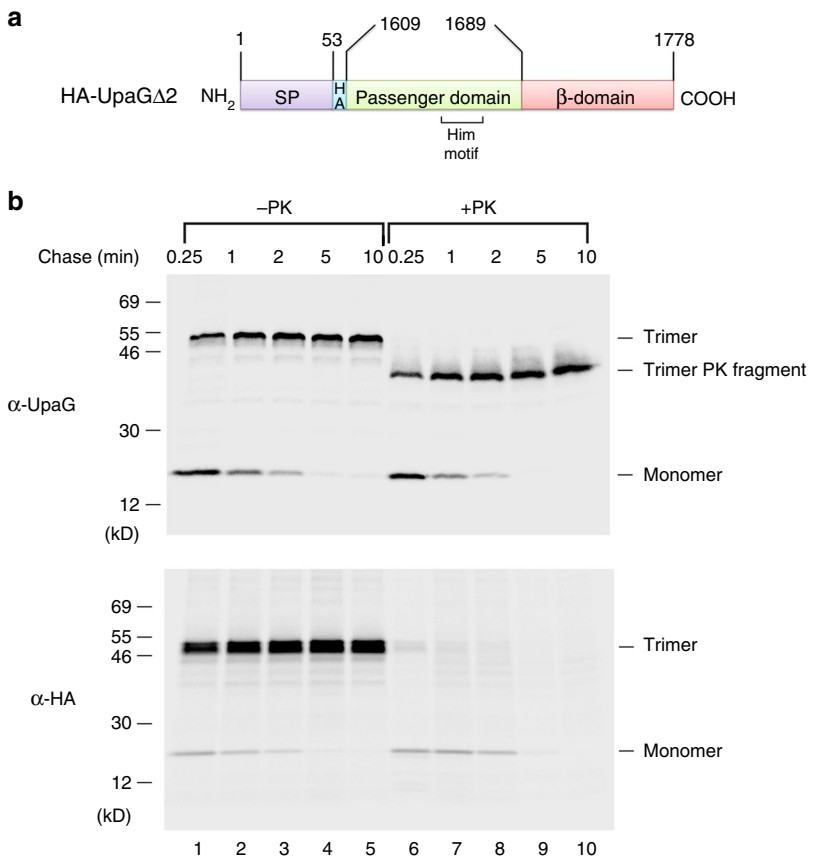

**Fig. 1** A model TAA assembles rapidly. **a** Illustration of HA-UpaGΔ2, an HA-tagged derivative of UpaG that contains the C-terminal 170 residues of the protein. The signal peptide (SP) (residues 1–53 of the full-length protein), the passenger-domain fragment (residues 1609–1689) and the segment that trimerizes to form the β-domain (residues 1689–1778) are indicated. A predicted Him motif (residues 1655–1675) is also shown. An HA tag was inserted at the N terminus of the passenger domain. **b** AD202 transformed with pRS1 (P$_{trc}$-HA-upaGΔ2) were subjected to pulse-chase labeling. Half of the cells were treated with PK, and immunoprecipitations were performed using an anti-UpaG or an anti-HA antiserum

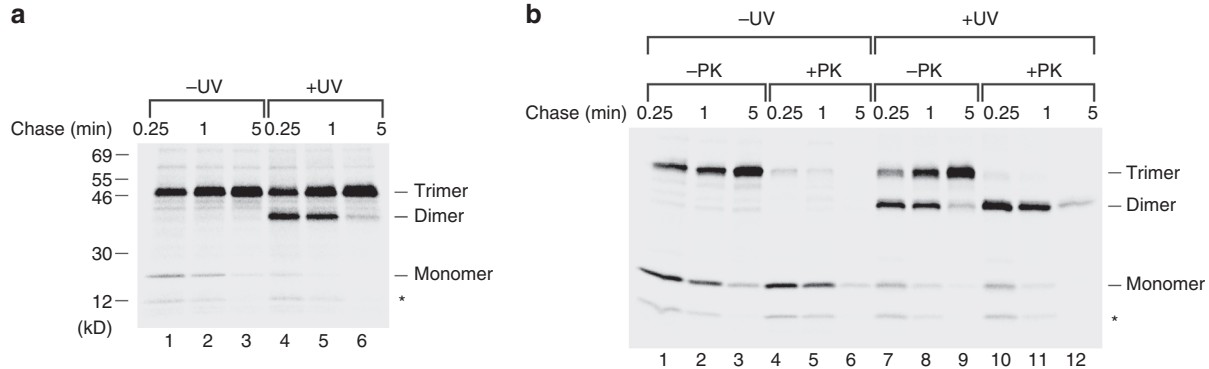

**Fig. 2** UpaGΔ2 monomers efficiently form a crosslinking product. **a** AD202 transformed with pRS4 [P$_{trc}$-HA-upaGΔ2(Y1735am)] and pDULE-Bpa were subjected to pulse-chase labeling. Half of the cells were UV irradiated, and immunoprecipitations were performed using an anti-HA antiserum. The truncated form of the protein that resulted from translation termination at the amber codon is denoted with an asterisk. **b** The experiment shown in (**a**) was repeated except that half of each sample was treated with PK

To gain insight into the mechanism of OMP assembly and the role of the Bam complex in this process, we decided to conduct a detailed analysis of the biogenesis of a protein produced by the uropathogenic *E. coli* strain CFT073 called UpaG that has a distinctive architecture[22, 23]. UpaG is a member of a family of virulence factors known as trimeric autotransporter adhesins (TAAs) that facilitate attachment of bacteria to host cells[24]. Like classical (monomeric) autotransporters, TAAs consist of two domains, an N-terminal extracellular ("passenger") domain and a C-terminal β-barrel domain that is integrated into the OM by the Bam complex[25]. TAAs differ from classical autotransporters, however, in that they are assembled from three identical subunits. Each subunit contributes four β-strands to a single 12-stranded β-barrel domain and one strand to a trimeric passenger domain that folds into alternating coiled-coil and β-prism or β-roll-type structures[26–29]. The two domains are connected by three α-helices (one contributed by each subunit) that traverse the β-barrel pore and form the base of the first coiled-coil region. The biogenesis of TAAs has been studied only superficially, and mutations that perturb assembly have been classified mainly by their effect on protein stability[25, 30–32]. Nevertheless, based on their similarity to classical autotransporters it seems likely that the passenger domain is secreted in a C-to-N-terminal fashion through the formation of a hairpin structure[33–35].

Because TAAs are formed from three unlinked subunits, they provide a unique experimental tool for monitoring β-barrel assembly. In this study, we combine pulse-chase labeling with site-specific photocrosslinking to obtain snapshots of UpaG assembly in vivo. By examining a wild-type version of UpaG and a mutant that fails to interact with the Bam complex, we find that the β-barrel assembles in the periplasm into an asymmetric trimer that persists even after the protein is integrated into the OM. Our results provide evidence that the Bam complex can engage a β-barrel protein that is targeted in a partially folded conformation. As a corollary, we identify several distinct stages in the assembly of a family of virulence factors that has not been characterized in detail.

## Results

**Rapid assembly of an N-terminally truncated form of UpaG.** Complete assembly of a TAA requires the formation of a trimeric structure, integration of the β-barrel domain into the OM, and secretion of the passenger domain. Previous studies have shown, however, that only the ~80 C-terminal residues that correspond to the β-barrel and the embedded α-helical linker segments are required to form a fully folded and properly localized trimer that

is resistant to heat and SDS denaturation[30, 31]. As a first step towards elucidating the UpaG assembly pathway, we wished to examine the kinetics of passenger-domain translocation and stable trimer formation. Presumably because UpaG promotes cell aggregation[22], we found that production of the full-length (1778 residue) protein was toxic. To circumvent this problem, we examined the assembly of N-terminally truncated derivatives. Portions of *upaG* were cloned into pTrc99a under the control of the *trc* promoter, and an epitope tag was inserted between the native signal peptide and the N terminus of the passenger-domain fragment to facilitate detection of the protein. We found that a 170 residue derivative of UpaG that contains ~80 residues of the native-passenger domain and an HA tag (HA-UpaGΔ2; Fig. 1a) is relatively non-toxic and highly amenable to experimental analysis. This derivative contains a single Him motif (residues 1655–1675) that is often associated with bacterial adhesins[22].

An examination of HA-UpaGΔ2 provided evidence that TAAs are assembled very rapidly. *E. coli* strain AD202 (MC4100 *ompT::kan*) transformed with a plasmid that encodes HA-UpaGΔ2 were grown in M9 minimal medium and subjected to pulse-chase labeling. Cells were collected by centrifugation and half of each sample was treated with proteinase K (PK) to assess the exposure of the passenger domain. Immunoprecipitations were then conducted using anti-UpaG and anti-HA antisera. By 15 s, a substantial fraction of the ~20 kD HA-UpaGΔ2 monomer was assembled into stable trimers that migrated at ~50 kD and were resistant to SDS and heat denaturation. Virtually all of the monomer was assembled within 2 min (Fig. 1b, lanes 1–5). It is important to note that because the antisera bind to the monomer and trimer with different affinities (the anti-HA antiserum in particular appeared to bind much more efficiently to the trimer), assembly can only be assessed by monitoring the accumulation of the trimer. Following treatment with PK, a ~42 kD polypeptide was detected with anti-UpaG but no oligomeric forms could be detected with anti-HA (Fig. 1b, lanes 6–10). The simplest explanation of this result is that while the N-terminal HA tags were exposed on the cell surface and removed by the protease, the Him motif was present on the cell surface as a folded structure that, together with the entire β-barrel, (~125 residues from each subunit) was protected from digestion.

**UpaG β-barrel assembly begins before passenger-domain export.** Our analysis showed that unlike the stable trimer, the monomeric form of HA-UpaGΔ2 was completely resistant to PK digestion (Fig. 1b, lanes 6–10). This observation provided the first indication that passenger-domain translocation occurs after the

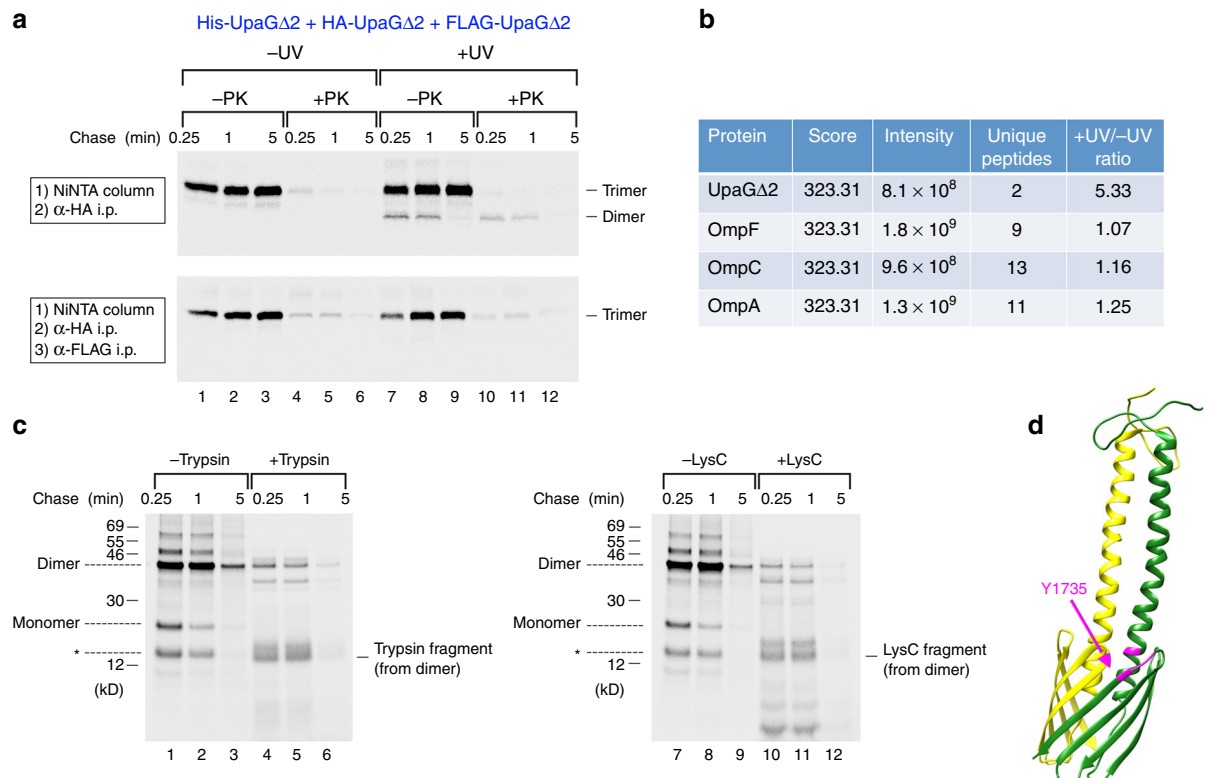

**Fig. 3** The UpaGΔ2 crosslinking product is a homodimer. **a** AD202 transformed with pRS9 [P$_{trc}$-*His$_{10}$-upaGΔ2(Y1735am)* + *HA-upaGΔ2(Y1735am)* + *FLAG-upaGΔ2(Y1735am)*] and pDULE-Bpa were subjected to pulse-chase labeling. Half of the cells were UV irradiated, and half of both the irradiated and non-irradiated samples were treated with PK. His$_{10}$-tagged proteins purified by Ni-NTA chromatography were then subjected to immunoprecipitation with an anti-HA antiserum (top gel) or sequential immunoprecipitations with anti-HA and anti-FLAG antisera (bottom gel). **b** AD202 were transformed with pRS5 [P$_{trc}$-*His$_{10}$-upaGΔ2(Y1735am)*], and half of the cells were UV irradiated. His$_{10}$-tagged proteins were purified from cell membranes by Ni-NTA chromatography and resolved by SDS-PAGE. Proteins in the ~40 kD range purified from both irradiated and non-irradiated were extracted, labeled with light and heavy isotopes, respectively, and analyzed by quantitative mass spectrometry. The MaxQuant score and intensity of each protein and its relative abundance in the two samples are shown. **c** AD202 transformed with pRS4 [P$_{trc}$-*HA-upaGΔ2(Y1735am)*] and pDULE-Bpa were radiolabeled, UV irradiated, and treated with PK. After proteins were TCA precipitated, immunoprecipitations were conducted using an anti-HA antiserum. One half of each sample was then treated with trypsin or Lys-C. **d** Structure of two subunits of Hia[27] visualized using UCSF Chimera software (http://www.rbvi.ucsf.edu/chimera). The conserved tyrosine (Y1055) that is equivalent to UpaG Y1735 in one subunit and all residues located within 4 Å in the adjacent subunit are shown in magenta

formation of a trimeric structure and disfavors a model in which translocation follows the membrane insertion of monomers that subsequently trimerize. To further examine the early stages of UpaG assembly, we used a site-specific photocrosslinking method in which the expression of an archaeal amino acyl-tRNA synthetase and amber suppressor tRNA pair facilitates the incorporation of the photoactivatable amino acid analog Bpa at an amber codon engineered into a protein of interest[36]. Upon photoactivation by UV light, Bpa is crosslinked to molecules that lie within ~4 Å of the polypeptide backbone. Amber mutations were introduced into HA-UpaGΔ2 at various positions, and AD202 were transformed with plasmids encoding an amber mutant and the amber suppression system (pDULE-Bpa). Cells were subjected to pulse-chase labeling, and half of the cells collected at each time point were irradiated with UV light. Immunoprecipitations were then performed using an anti-HA antiserum.

We found that when Bpa was introduced into HA-UpaGΔ2 at position 1735 in place of tyrosine, the monomer almost completely disappeared and a prominent crosslinking product that has the same predicted molecular weight as a dimer (~40 kD) was observed upon UV irradiation (Fig. 2a, lanes 4–6). Although a crosslink between the monomer and the ~17 kD Skp chaperone

might also produce a ~40 kD band, the same crosslinking pattern was observed in a Δ*skp* strain (Supplementary Fig. 1). The exceptional efficiency of the crosslinking reaction implies that the product results from a highly stable and uniform intermolecular interaction. A time-dependent and temperature-dependent decrease in the level of the ~40 kD band that paralleled an increase in the level of the stable trimer also provides evidence that it corresponds to an assembly intermediate (Fig. 2a, lanes 4–6; Supplementary Fig. 2, lanes 7–9). Interestingly, the crosslinking product was completely resistant to PK digestion and therefore must result from an interaction that occurs in the periplasmic space prior to the initiation of passenger-domain translocation (Fig. 2b, lanes 10–12; Supplementary Fig. 2, lanes 10–12).

Further analysis showed that the crosslinking product corresponds to a covalently linked HA-UpaGΔ2 dimer. To test the possibility that the ~40 kD polypeptide contains more than one UpaGΔ2 subunit, we created a plasmid that encodes HA-tagged, His$_{10}$-tagged, and FLAG-tagged UpaGΔ2(Y1735am) under the control of a single *trc* promoter after we found that His$_{10}$-tagged and FLAG-tagged forms of UpaGΔ2 assemble as rapidly as HA-UpaGΔ2 (Supplementary Fig. 3). By producing the three differentially tagged forms of the protein simultaneously, we

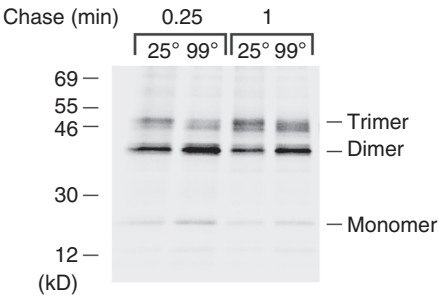

**Fig. 4** The crosslinked UpaGΔ2 dimer is derived from an unstable trimer. AD202 transformed with pRS5 [P$_{trc}$-His$_{10}$-upaGΔ2(Y1735am)] and pDULE-Bpa were radiolabeled, UV irradiated, and treated with PK. Following cell lysis and solubilization of cell membranes in DDM, His$_{10}$-tagged proteins were purified by Ni-NTA chromatography and incubated at 25 or 99 °C in SDS sample buffer prior to SDS-PAGE

could use a sequential purification approach to determine whether the crosslinking product contains multiple UpaGΔ2 subunits. AD202 transformed with the plasmid and pDULE-Bpa were subjected to pulse-chase labeling, and half of the cells collected at each time point were UV irradiated. Samples were then divided in half and one portion was treated with PK. Both monomeric and oligomeric forms of UpaGΔ2 that contain at least one His$_{10}$-tagged subunit were purified by Ni-NTA chromatography. The protein eluted from the column was then subjected to immunoprecipitation with an anti-HA antiserum or two sequential immunoprecipitations with anti-HA and anti-FLAG antisera. Both stable trimers and ~40 kD crosslinked molecules that contain His$_{10}$-tagged and HA-tagged subunits were immunoprecipitated by the anti-HA antiserum alone (Fig. 3a, top, lanes 7–12). In contrast, only stable trimers were isolated when sequential immunoprecipitations were performed (Fig. 3a, bottom). These results indicate that the ~40 kD band is composed of two covalently crosslinked subunits of UpaGΔ2 and rule out the possibility that it contains three subunits.

Mass spectrometry confirmed that the crosslinking product consists entirely of His$_{10}$-UpaGΔ2 subunits. AD202 were transformed with a plasmid encoding His$_{10}$-UpaGΔ2(Y1735am) and pDULE-Bpa, and the TAA was overproduced in a large-scale culture. Half of the cells were UV irradiated, and cell membranes were isolated from both treated and untreated cells. His$_{10}$-UpaGΔ2 monomers and oligomers were then purified on a Ni-NTA column and resolved by SDS-PAGE. Owing to an unavoidable contamination of the samples with abundant OMPs (which were present in great excess over His$_{10}$-UpaGΔ2), the crosslinking product was not well resolved after Colloidal Blue staining (Supplementary Fig. 4A). Nevertheless, a western blot performed with the anti-UpaG antiserum showed that a ~40 kD polypeptide was present primarily in the UV-irradiated sample (Supplementary Fig. 4B). The ~40 kD region of the stained gel was excised from both the –UV and +UV lanes and proteins extracted from the gel slices were labeled with light and heavy isotopes, respectively. Tryptic fragments were then analyzed by quantitative mass spectrometry. The only high-intensity peptides that predominated in the +UV sample (+UV/–UV ratio = 5.33) were derived from UpaGΔ2 (Fig. 3b; Supplementary Datasets 1–2).

To identify the site of interaction between two adjacent UpaGΔ2 subunits that leads to the formation of the crosslinking product, we exploited the fortuitous location of residue 1735 in large tryptic and Lys-C fragments (50 and 54 residues, respectively) (Supplementary Fig. 5A). AD202 transformed with the plasmid that encodes HA-UpaGΔ2(Y1735am) and pDULE-

Bpa were subjected to pulse-chase labeling and photocrosslinking. Intact cells were treated with PK to remove the surface exposed HA-tagged passenger domains from stable trimers, proteins were TCA precipitated, and immunoprecipitations were performed using the anti-HA antiserum. Half of the immunoprecipitated protein was then digested with either trypsin or Lys-C. As expected, the crosslinked HA-UpaGΔ2 dimer was the major species that was isolated following PK treatment alone (Fig. 3c, lanes 1–3 and 7–9). Subsequent treatment with trypsin or Lys-C led to the degradation of the dimer and the appearance of a polypeptide that migrated slightly >12 kD (Fig. 3c, lanes 4–6 and 10–12). The large size of these tryptic/Lys-C fragments strongly suggests that they contain two crosslinked 50 (or 54) residue proteolytic fragments derived from two adjacent HA-UpaGΔ2 subunits (predicted molecular weight = 11–12 kD). Control experiments confirmed that these peptides were only observed upon UV irradiation and were derived from crosslinked dimers rather than residual stable trimers (Supplementary Figs. 6 and 7). Indeed an ~11 kD tryptic peptide was presumably not detected by mass spectrometry because it was too large to diffuse readily out of polyacrylamide gel slices.

The tyrosine at position 1735 of UpaG is highly conserved, and crystal structures of the fully folded C terminus of Hia, a *Haemophilus influenza* TAA[27], and YadA, a more distantly related *Yersinia enterocolitica* TAA[28], show that the equivalent tyrosine is within 4 Å of several residues in the adjacent subunit (Fig. 3d; Supplementary Fig. 8). Interestingly, some of these target residues are located in a region that corresponds to the N terminus of the 50 amino acid tryptic fragment of UpaGΔ2 (Supplementary Fig. 5B). The formation of a covalent bond between Bpa and one of these residues might sterically impede the access of proteases to the upstream lysine(s) and explain why the tryptic/Lys-C fragments observed in Fig. 3c appear to be slightly larger than expected. In any case, taken together the structural and crosslinking data raise the possibility that the ~40 kD dimer corresponds to an assembly intermediate in which two interacting subunits adopt a conformation that resembles their conformation in the final folded structure.

Because it seems unlikely that a homotrimer would form from a dimeric intermediate, we hypothesized that the ~40 kD crosslinking product is part of an unstable, asymmetric trimer. In this trimeric intermediate, the Bpa molecule introduced at residue 1735 of one subunit would be located close enough to another subunit to form a covalent bond after UV irradiation, but the third subunit would dissociate from the complex during sample preparation or upon heating in SDS. Although we could not detect trimers by Blue Native PAGE or chemical crosslinking, we obtained evidence for a trimeric intermediate using another approach. AD202 transformed with a plasmid encoding His$_{10}$-UpaGΔ2(Y1735am) and pDULE-Bpa were subjected to pulse-chase labeling and photocrosslinking. Intact cells were then treated with PK to remove the surface exposed His-tagged passenger domains from stable trimers. Proteins were solubilized from isolated cell membranes using detergent, and His-tagged monomers and oligomers were purified on a Ni-NTA column. The eluate from the column was then incubated in SDS sample buffer at either 25 or 99 °C and resolved by SDS-PAGE. Consistent with our hypothesis, we observed a significantly higher level of a trimeric form of the protein in the unheated sample (Fig. 4). We also saw a concomitant increase in the level of dimers and monomers in the heated sample that presumably resulted from the dissociation of the trimer. Taken together with our analysis of the crosslinking product, these results suggest that three UpaGΔ2 subunits fold into an asymmetrical (and potentially open) β-barrel-like structure prior to the initiation of translocation.

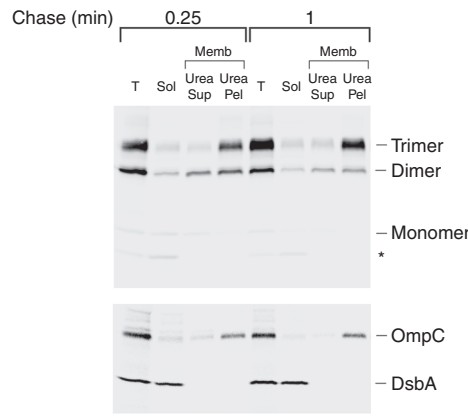

**Fig. 5** Localization of oligomeric forms of UpaGΔ2. AD202 transformed with pRS4 [P*trc*-HA-upaGΔ2(Y1735am)] and pDULE-Bpa were radiolabeled, UV irradiated, and fractionated. Immunoprecipitations were conducted using an anti-HA antiserum (top gel) or anti-OmpC and anti-DsbA antisera (bottom gel). The truncated form of HA-UpaGΔ2 that resulted from translation termination at the amber codon is denoted with an asterisk. T, total cell extract; Sol, soluble protein; Memb, cell membranes; Urea Sup, urea-soluble protein; Urea Pel, urea-insoluble protein

**Assembly of the UpaG β-barrel is initiated in the periplasm.** We next performed cell fractionation experiments to dissect the UpaGΔ2 assembly pathway further. AD202 transformed with a plasmid encoding HA-UpaGΔ2(Y1735am) and pDULE-Bpa were subjected to pulse-chase labeling and photocrosslinking. Cell lysates were separated into soluble and membrane fractions, and the membranes were incubated with 6 M urea to remove loosely associated proteins. The urea-washed membranes were then isolated by centrifugation. To validate the fractionation, we performed immunoprecipitations using antisera against an OMP (OmpC) and a periplasmic protein (DsbA). As expected, OmpC was located in the membrane fraction and was completely resistant to urea extraction while DsbA was located exclusively in the soluble fraction (Fig. 5, bottom). We then performed immunoprecipitations using the anti-HA antiserum. Like OmpC, essentially all of the stable UpaGΔ2 trimer was membrane-associated and resistant to urea extraction. In contrast, the crosslinked dimer was found in all of the fractions. Roughly equal amounts of the dimer were located in the integral membrane and peripheral membrane fractions, and a small but significant amount was located in the soluble fraction (Fig. 5, top; a darker exposure is shown in Supplementary Fig. 9). The identification of a large population of urea-extractable dimers confirms that the assembly of UpaGΔ2 begins prior to its membrane integration. Given that the dimer is completely resistant to PK digestion, however, the detection of a second non-extractable population implies the existence of a late pre-translocation assembly intermediate in which the overall architecture is maintained but the β-barrel is integrated into the OM. The finding that a significant portion of the UpaGΔ2 monomer co-fractionates with the dimer in the membrane fractions, whereas the unsuppressed amber fragment is located entirely in the soluble fraction (Supplementary Fig. 9) provides further evidence that the dimer represents a component of an unstable trimer. Finally, the detection of the crosslinked dimer in the soluble fraction is particularly striking because it suggests that the assembly of the UpaG β-barrel begins in the periplasm.

We subsequently used an UpaGΔ2 mutant that has a targeting defect to determine whether TAA assembly is initiated prior to its interaction with the Bam complex. Like most bacterial OMPs, TAAs contain a C-terminal "β-signal" motif that includes an aromatic C-terminal amino acid[32]. The mutation of residues in this motif inhibits OMP assembly[25, 37–40]. Several studies, including a study on YadA, have indicated that OMPs require the motif to bind to the Bam complex[25, 39]. Although we found that the mutation of the C-terminal tryptophan residue of HA-UpaGΔ2 (residue 1778) to alanine strongly impaired assembly, the effect was conditional. The mutant protein was assembled into stable trimers in cells that were grown in M9, but only at a relatively slow rate (Supplementary Fig. 10). Furthermore, the presence of low molecular weight bands that disappeared over time suggested that some of the monomer remained unassembled and was degraded. In contrast, no assembly was observed when the cells were grown in LB (Supplementary Fig. 11). These results suggest that the assembly defect correlates with the rate of cell growth. Presumably slow growth conditions reduce the defect by increasing the time window during which the mutant protein can interact productively with the Bam complex. Indeed, slow growth conditions have previously been shown to create a permissive condition for the assembly of a non-native OMP[41].

Because we wished to radiolabel HA-UpaGΔ2(W1778A) and monitor the fate of newly synthesized protein under conditions that enhance the assembly defect, we next grew cells in modified EZ Rich medium[42], an enriched defined medium that stimulates faster growth than M9. Curiously, we found that the production of low levels of the mutant protein due to the leakiness of the *trc* promoter was toxic in EZ Rich medium in experiments that required transforming cells with two plasmids. For this reason, we cloned the gene encoding wild-type or mutant HA-UpaGΔ2 into pSCRhaB2 under the control of a tightly regulated *rhaB* promoter. Initially, AD202 transformed with one of the plasmids were subjected to pulse-chase labeling following the addition of 0.2% rhamnose. Immunoprecipitations were then performed using the anti-HA antiserum. Whereas wild-type HA-UpaGΔ2 was assembled rapidly, essentially no stable HA-UpaGΔ2 (W1778A) trimers were observed and the monomeric form of the protein was gradually degraded (Fig. 6a). Despite the instability of the mutant protein, we detected a crosslinked dimer at all time points when we introduced Bpa at position 1735 under the same growth conditions and UV irradiated the samples (Fig. 6b, lanes 10–12). The dimer was neither degraded nor converted into a stable trimer after a 5 min chase. These results strongly suggest that under relatively rapid growth conditions UpaGΔ2 (W1778A) molecules begin to fold but cannot integrate into the OM because they fail to interact productively with the Bam complex.

To obtain additional evidence that HA-UpaGΔ2(W1778A) oligomerizes in the periplasm, cells were grown in EZ Rich medium, subjected to pulse-chase labeling and UV irradiation, and separated into soluble and membrane fractions. The membrane fraction was then incubated with 6 M urea to extract peripherally bound proteins. Interestingly, the crosslinked dimer was almost equally distributed between the soluble and urea-extractable membrane protein fractions even after a 5 min chase (Fig. 7). None of the protein was detected in the urea-resistant membrane protein fraction. The finding that the HA-UpaGΔ2 (W1778A) monomer co-fractionated with the crosslinked dimer is consistent with our other evidence that the protein assembles into an asymmetric trimer that dissociates when samples are heated in SDS sample buffer. In any case, the presence of a large percentage of the dimer in the soluble fraction and the persistence of the fractionation pattern over time strongly suggest that the mutant protein begins to fold but effectively remains trapped in the periplasm.

Finally, we conjectured that if the W1778A mutation affects the targeting of a barrel-like structure formed in the periplasm to the OM, then the assembly defect might be suppressed by the

incorporation of the mutant protein into mixed trimers that contain at least one wild-type subunit. To test this idea, we cloned genes encoding the HA-UpaGΔ2(W1778A) mutant and wild-

type His$_{10}$-UpaGΔ2 into a single plasmid behind a common *rhaB* promoter. As a control, we constructed an analogous plasmid encoding both HA-tagged and His$_{10}$-tagged versions of UpaGΔ2

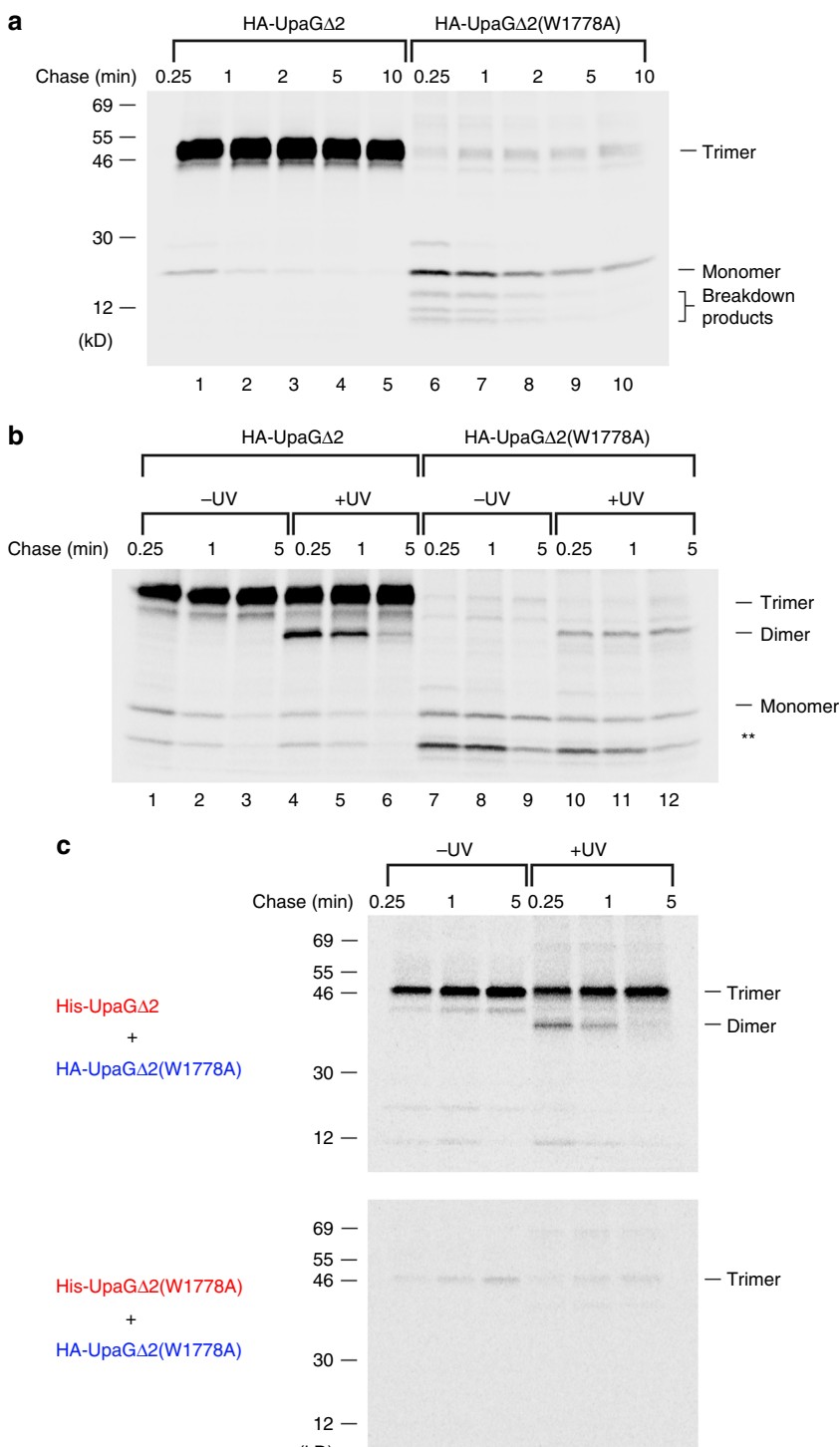

**Fig. 6** Assembly of a targeting-deficient UpaGΔ2 mutant under enhanced growth conditions. **a** AD202 transformed with pRS10 [P$_{rha}$-*HA-upaGΔ2*] or pRS12 [P$_{rha}$-*HA-upaGΔ2(W1778A)*] were grown in in EZ Rich medium and subjected to pulse-chase labeling. Immunoprecipitations were then conducted using an anti-HA antiserum. **b** AD202 transformed with pDULE-Bpa and either pRS11 [P$_{rha}$-*HA-upaGΔ2(Y1135am)*] or pRS13 [P$_{rha}$-*HA-upaGΔ2(Y1135am/W1778A)*] were radiolabeled and half of each sample was UV irradiated. Immunoprecipitations were then conducted using an anti-HA antiserum. The truncated form of HA-UpaGΔ2 that resulted from translation termination at the amber codon and a possible breakdown product of HA-UpaGΔ2(W1778A) are denoted with a double asterisk. **c** AD202 transformed with pDULE-Bpa and either pRS16 [P$_{rha}$-*HA-upaGΔ2(Y1135am/W1778A)* + *His$_{10}$-upaGΔ2(Y1135am)*] or pRS17 [P$_{rha}$-*HA-upaGΔ2(Y1135am/W1778A)* + *His$_{10}$-upaGΔ2(Y1135am/W1778A)*] were radiolabeled and half of each sample was UV irradiated. Proteins eluted from a Ni-NTA column were then subjected to Immunoprecipitations using an anti-HA antiserum

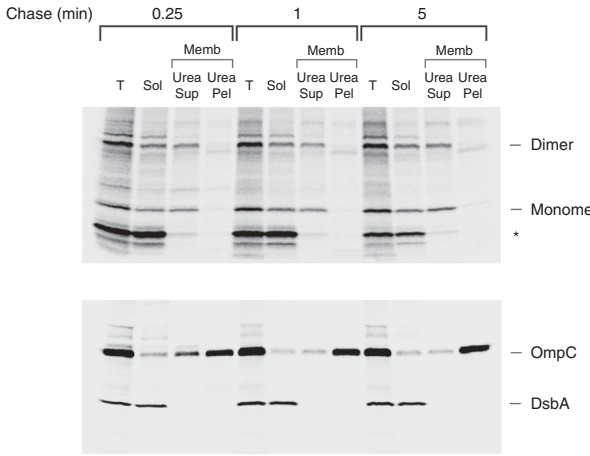

**Fig. 7** Localization of the UpaGΔ2(W1778A) crosslinking product. AD202 transformed with pRS13 [P_rha-HA-upaGΔ2(Y1135am/W1778A)] and pDULE-Bpa were radiolabeled, UV irradiated and fractionated. Immunoprecipitations were conducted using an anti-HA antiserum (top gel) or anti-OmpC and anti-DsbA antisera (bottom gel). The truncated form of HA-UpaGΔ2 that resulted from translation termination at the amber codon is denoted with an asterisk. T, total cell extract; Sol, soluble protein; Memb, cell membranes; Urea Sup, urea-soluble protein; Urea Pel, urea-insoluble protein

(W1778A). Cells were grown in EZ Rich medium and subjected to pulse-chase labeling after the addition of 0.2% rhamnose. Half of each sample was then UV irradiated. Oligomers containing both His-tagged and HA-tagged subunits were purified by Ni-NTA chromatography followed by immunoprecipitation with the anti-HA antiserum. Consistent with our hypothesis, we observed the formation of both crosslinked dimers and stable heterotrimers containing wild-type His$_{10}$-UpaGΔ2 and HA-UpaGΔ2(W1778A) subunits (Fig. 6c). As expected, the two different tagged versions of UpaGΔ2(W1778A) did not form stable trimers effectively. These results strongly suggest that wild-type UpaGΔ2 compensates for the targeting defect caused by the W1778A mutation by interacting with the mutant protein and forming an assembly-competent intermediate in the periplasm. Furthermore, the results demonstrate that the mutation does not compromise the intrinsic ability of the protein to fold into a native conformation.

## Discussion

By using site-specific photocrosslinking to monitor relatively long-lived intermolecular interactions between the three identical subunits that form the β-barrel domain of the TAA derivative UpaGΔ2, we obtained both temporal and spatial information about its assembly. The rapid appearance of an extremely abundant crosslinking product that corresponds to a covalently linked dimer and the concomitant disappearance of the monomeric form of the protein shows that the protein assembles efficiently into a uniform structure soon after its translocation across the IM. SDS-PAGE analysis and cell fractionation experiments provided evidence that the crosslinked dimer is derived from (and can be used as a proxy for) an unstable trimeric assembly intermediate. In any case, the observation that ~50% of the crosslinking product could be extracted from cell membranes with urea clearly demonstrates that assembly is initiated prior to membrane integration. These results are difficult to reconcile with a model in which TAA β-barrels are assembled through the stepwise threading of individual β-strands through the BamA lateral gate.

Experiments performed with an UpaGΔ2 variant that has a mutation in a critical targeting motif provide strong evidence that assembly is initiated in the periplasm. Like the wild-type protein, the UpaGΔ2(W1778A) mutant formed a dimer upon UV irradiation, but about half of the crosslinking product was located in the soluble fraction. The observation that the fractionation pattern persisted over time and that a small amount of the wild-type UpaGΔ2 crosslinking product was also found in the soluble fraction suggests that the mutation prolongs an early stage of the normal assembly pathway. Although the remainder of the mutant dimer co-localized with peripheral membrane proteins, previous results[43] suggest that this might be a soluble population that associates non-specifically with cell membranes or aggregates during the fractionation procedure. In addition to the cell fractionation results, the conditional nature of the UpaGΔ2 (W1778A) assembly defect is consistent with the idea that the mutation impairs the binding to BamA. The simplest interpretation of the data is that slow growth conditions increase the time window during which the mutant protein can interact productively with the Bam complex rather than misfold. It is conceivable that the mutation affects a post-targeting assembly step, but the finding that the incorporation of a wild-type UpaGΔ2 subunit into mixed oligomers suppresses the assembly defect indicates that the mutation does not affect protein folding per se.

In addition to providing insight into the folded state of an OMP before it interacts with the Bam complex, our analysis of UpaGΔ2 enabled us to construct a detailed working model for TAA assembly. Our results suggest that UpaGΔ2 folds rapidly in the periplasm into an asymmetric trimer in which the spatial arrangement of the two crosslinked subunits resembles that observed in the fully folded structure (Fig. 8, step 1). The third subunit interacts with the other two subunits in a different fashion, but binds tightly enough to remain associated in the presence of SDS unless heated. We propose that at this stage all three linker segments are embedded inside of a barrel-like structure. The linker segment, which contains a potentially flexible region located near the extracellular side of the β-barrel[28], might form a hairpin with the passenger domain that facilitates translocation and prevents the barrel from closing until the translocation reaction is complete. Indeed the finding that the linker segment is essential for TAA assembly[30, 31] suggests that it might nucleate trimerization. Although our results show that UpaGΔ2 begins to fold very rapidly, the persistence of the crosslinked dimer and its resistance to PK digestion indicates that there is at least one slow step between the formation of the asymmetric trimer and the initiation of passenger-domain translocation. Interestingly, the discovery of a large population of crosslinked dimers that is resistant to urea extraction and PK digestion implies that the passenger domain is not immediately exposed on the cell surface after the C terminus is integrated into the OM. The existence of this novel assembly intermediate implies that after the β-barrel domain is targeted to the Bam complex and inserted into the OM (Fig. 8, step 2) a transition occurs that triggers the onset of translocation (Fig. 8, step 3). The ability of wild-type UpaGΔ2 to suppress the assembly defect caused by the W1778A mutation suggests that a single functional β-signal motif may be sufficient to mediate the targeting reaction itself.

Our finding that both the surface exposure of the UpaGΔ2 passenger domain and the accumulation of SDS and heat-resistant trimers occur within seconds shows that translocation is very rapid (Fig. 8, step 4). In this regard it should be noted that both truncated and full-length passenger domains of classical autotransporters are secreted very rapidly[33, 41]. Because we did not observe crosslinking of the UpaGΔ2 β-barrel domain to Bam

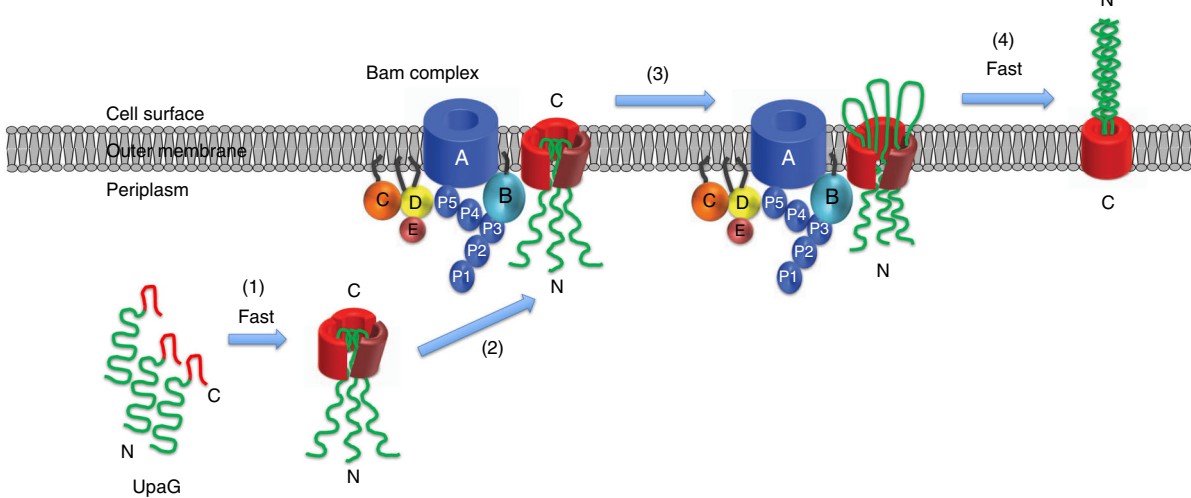

**Fig. 8** Model for the assembly of a TAA. Our results suggest that after a TAA is translocated into the periplasm, three subunits rapidly form an asymmetric trimer in which two subunits fold into a structure that reflects their position in the final structure (step 1). Subsequently, this trimeric intermediate is targeted to the Bam complex and integrated into the OM (step 2). The initiation of passenger-domain translocation is then triggered by a relatively slow transition (step 3). After the passenger domain is rapidly translocated across the OM the β-barrel collapses into a heat-resistant and SDS-resistant structure (step 4)

complex subunits when we introduced Bpa at a variety of positions, we cannot determine whether translocation occurs, whereas the protein is associated with the Bam complex. Indeed the discovery of a membrane-embedded pre-translocation intermediate is consistent with the proposal that TAA β-barrel domains dissociate from the Bam complex and catalyze passenger-domain translocation autonomously[35]. Given the apparent instability of this intermediate that may inhibit its partitioning into the lipid bilayer and considerable evidence that classical autotransporters are not self-contained secretion systems[44], however, we favor the view that translocation requires either the direct or indirect participation of the Bam complex.

Although our results suggest that TAAs trimerize in the periplasm and that, as a corollary, the Bam complex can mediate the insertion of at least partially folded client proteins, it should be of great interest to determine if some OMPs are assembled by alternative mechanisms. Several lines of evidence indicate that classical autotransporters likewise begin to fold prior to their membrane integration[41, 43, 45, 46]. Furthermore, a putative late-stage assembly intermediate was recently described in which the OMP LptD is folded around a lipoprotein plug (LptE) while still bound to the Bam complex[47]. The membrane integration of the BamA β-barrel itself has been shown to require the burial of an internal loop[40], and there is evidence that analogous interactions between a loop and residues in the barrel wall may also occur before PhoE is inserted into the OM[48]. It is unclear if the Bam complex has a critical role in promoting the folding of the proteins that were analyzed in these studies, but the results provide further evidence that β-barrel proteins that contain an embedded polypeptide segment begin to fold prior to their membrane integration. Given that the β-barrels of many OMPs are empty, however, it is conceivable that proteins like autotransporters and LptD represent special cases. Indeed the acquisition of a native structure may require that embedded polypeptides are incorporated into a barrel-like structure at an early stage of assembly and thereby impose constraints on the assembly mechanism. In contrast, OMPs that contain empty β-barrels may often arrive at the OM in an unfolded conformation. These proteins might then be inserted into the OM in a stepwise fashion through the BamA lateral gate. Consistent with the possibility that there are multiple OMP assembly pathways, different OMPs show differential sensitivity to the partial or complete loss of Bam complex subunits and periplasmic chaperones[49–51].

## Methods

**Reagents, bacterial strains, and growth conditions**. The *E. coli* K-12 strain AD202 (MC4100 *ompT::kan*)[52] was used in all experiments. Unless otherwise noted, all cultures were grown at 37 °C in M9 minimal medium containing 0.2% glycerol and all L-amino acids (40 μg ml$^{-1}$) except for methionine and cysteine or EZ Rich-defined medium containing 0.4% glycerol without methionine and cysteine (Teknova). Ampicillin (100 μg ml$^{-1}$), tetracycline (5 μg ml$^{-1}$), kanamycin (30 μg ml$^{-1}$), or trimethoprim (50 μg ml$^{-1}$) were added to the medium as necessary. A rabbit polyclonal antiserum was raised against purified UpaGΔ2. An antiserum raised against OmpC was described previously[53]. Rabbit polyclonal antisera against the HA epitope tag were obtained from Santa Cruz Biotechnology (catalog number sc-805) or Clontech (catalog number 631207). Mouse monoclonal antibodies recognizing the FLAG epitope tag (FG4R) (catalog number MA1-91878) was obtained from Thermo Fisher, respectively, and a rabbit anti-DsbA monoclonal antibody was obtained from Stressgen (catalog number SPA-659).

**Plasmid construction**. To construct a derivative of pTrc99a[54] that encodes UpaGΔ2 (pJH200), a fragment of *upaG* was amplified by PCR using primers JH0001 and JH0002 and *E. coli* O6:H1 strain CFT073 genomic DNA (ATCC 700928) as a template (all oligonucleotides are listed in Supplementary Table 1). The PCR product was then digested with *Eag*I and *Hin*dIII and cloned into the cognate sites of pJH36[55]. N-terminal HA and His$_{10}$ tags were attached to UpaGΔ2 by cloning oligonucleotide pairs JH0003/JH0004 and JH0005/JH0006 into the *Eag*I site of pJH200 to create pJH201 and pJH202, respectively. Plasmid pJH201 was further modified to create pJH203. In this plasmid, two glycine residues were inserted between the signal peptide and the HA tag using oligonucleotides JH0007 and JH0008 and the QuikChange Mutagenesis kit (Agilent) to improve the efficiency of signal peptide cleavage. The native UpaG signal peptide was then amplified by PCR using the primers PRS1001 and PRS1002 and *E. coli* CFT073 genomic DNA and cloned into the *Eco*RI and *Eag*I sites of pJH203 and pJH202 to replace the OmpA signal peptide and to create pRS1 and pRS2. To make a plasmid that encodes a FLAG-tagged version of UpaGΔ2 (pRS3), *upaGΔ2* was amplified by PCR using primers PRS1007 and PRS1008 and pRS1 as a template and cloned into the *Xba*I and *Hin*dIII sites of pTrc99a. The native signal peptide of UpaG was then amplified by PCR as described above and cloned into the *Eco*RI and *Eag*I sites of the resulting plasmid. The UpaG Y1735 codon was replaced with an amber codon in pRS1, pRS2, and pRS3 using primers PRS1003 and PRS1004 to create pRS4, pRS5, and pRS6, respectively. These and all subsequent mutations were produced using a previously described protocol[56]. A W1778A mutation was introduced into pRS1, pRS4, and pRS5, respectively, to create pRS7, pRS8, and pRS15 using primers PRS1012 and PRS1013. To generate a plasmid that expresses HA-tagged, His$_{10}$-tagged, and FLAG-tagged versions of UpaGΔ2(Y1735am) simultaneously (pRS9), His$_{10}$-UpaGΔ2(Y1735am) was first amplified by PCR using primers PRS1009 and PRS1010 and pRS5 as a template and cloned into the *Eco*RI site of pRS4. Subsequently, FLAG-UpaGΔ2(Y1735am) was amplified by PCR using primers PRS1011

and PRS1008 and pRS6 as a template and cloned into the *Hin*dIII site of the resulting plasmid.

To construct a plasmid that encodes HA-UpaGΔ2 under the control of the *rhaB* promoter (pRS10), HA-UpaGΔ2 was amplified by PCR using primers PRS1009 and PRS1008 and pRS1 as a template and cloned into the *Nde*I and *Hin*dIII sites of pSCRhaB2[57]. Derivatives that contain a Y1735am mutation (pRS11), a W1778A mutation (pRS12), and both Y1735am and W1778 mutations (pRS13) were made in an analogous fashion using pRS4, pRS7, and pRS8 as PCR templates. Plasmids pRS16 and pRS17 encode both HA-UpaGΔ2(Y1735/W1778A) and either $His_{10}$-UpaGΔ2(Y1735am) or $His_{10}$-UpaGΔ2(Y1735/W1778A) under the control of a single *rhaB* promoter and identical ribosome binding sites. They were constructed by first amplifying the gene encoding the $His_{10}$-tagged UpaGΔ2 variant using pRS5 or pRS15 as a template and primers PRS1014 and PRS1008. The PCR product was then cloned into the *Hin*dIII site of pRS13. Finally, to make pRS14, UpaGΔ2 was amplified by PCR using primers PRS1005 and PRS1006 and pRS1 as a template and cloned into the *Nde*I and *Bam*HI sites of pET28b (Novagen).

**Overproduction and purification of UpaGΔ2.** *E. coli* strain BL21(DE3) (Thermo Fisher) was transformed with plasmid pRS14. Overnight cultures were diluted 1:100 into 600 ml LB-containing kanamycin. Cultures were grown at 37 °C to $OD_{600}$ ~ 0.6 and production of His-tagged UpaGΔ2 was induced by the addition of 1 mM IPTG. After 3 h the cells were collected by centrifugation (3000×*g*, 20 min, 4 °C). The cell pellet was resuspended in 50 mM Tris pH 8, 5 mM EDTA, 2 mM PMSF, and lysed using an Emulsiflex-C3 instrument (Avestin). Inclusion bodies (which contained the UpaGΔ2 protein) were isolated by centrifugation (20,000×*g*, 5 min, 4 °C) and resuspended in 5 ml Bugbuster Master Mix (Novagen) and 1× Halt protease inhibitor cocktail (Thermo Fisher) for 30 min at room temperature to remove cell debris and membrane proteins. Following three washes with 0.1× Bugbuster Master Mix the inclusion bodies were solubilized in 100 mM $NaH_2PO_4$, 10 mM Tris pH 8, 8 M urea for 1 h at 4 °C. Insoluble matter was then removed by centrifugation (10,000×*g*, 30 min, 4 °C) and the supernatant was loaded onto a pre-equilibrated Ni-NTA agarose (Qiagen) column. His-tagged protein was purified according to the manufacturer's instructions for a non-native purification. The eluate from the Ni-NTA column was concentrated using an Amicon Ultra4 Centrifugal Filter Unit (EMD Millipore, 3 kDa cutoff). The concentrated protein was subjected to SDS-PAGE on Novex 8–16% Tris-glycine gels (Thermo Fisher Scientific) and bands corresponding to His-tagged UpaGΔ2 were excised and used for the production of a rabbit antiserum.

**Radiolabeling, photocrosslinking, and immunoprecipitations.** For experiments in which cells transformed with pTrc99a derivatives were grown in M9, overnight cultures were washed and diluted in fresh medium to $OD_{550} = 0.02–0.03$. Cultures were grown to $OD_{550} = 0.2–0.3$ and cells were subjected to pulse-chase labeling 30 min later as previously described[33]. No IPTG was added unless otherwise noted because radiolabeled protein produced as a result of residual transcription from the *trc* promoter was readily detected. For low-temperature experiments, cultures were shifted to 25 °C for 15 min prior to radiolabeling. For experiments in which cells transformed with pSCRhaB2 derivatives were grown in EZ Rich medium, overnight cultures were washed and diluted 1:100 and grown to $OD_{600} = 0.3$. L(+)-rhamnose (0.2% w/v) was then added to induce protein expression for 30 min prior to radiolabeling. All radiolabeled samples were collected by centrifugation (3000×*g*, 10 min, 4 °C) and resuspended in 1 ml M9 salts. In some experiments, the resuspended cells were divided in half, and one half was treated with 200 µg ml$^{-1}$ PK on ice for 20 min. The protease reaction was stopped by the addition of 2 mM phenylmethylsulfonyl fluoride (PMSF). Photocrosslinking was performed essentially as described except that 4 ml samples were obtained at each time point unless otherwise noted[33, 58]. In all experiments, 10% (w/v) trichloroacetic acid (TCA) was added to both treated and untreated samples to precipitate proteins. Immuno-precipitations were performed as previously described[59] and proteins were heated to 99 °C for 5 min unless otherwise noted and resolved by SDS-PAGE on Novex 8–16% Tris-glycine minigels (Thermo Fisher). Radiolabelled proteins were visualized using a Fujifilm FLA-9000 phosphorimager.

**Sequential purification of multiply-tagged UpaGΔ2 oligomers.** AD202 harboring plasmids pDULE-Bpa[58] and pRS9, pRS16 or pRS17 were radiolabeled and subjected to photocrosslinking as described above, except that 5 ml aliquots were obtained at each time point. In experiments that involved the simultaneous production of HA-tagged, $His_{10}$-tagged, and FLAG-tagged versions of UpaG (Y1735am), proteins were precipitated with TCA after half of each sample was treated with PK. TCA pellets were solubilized in buffer A (200 mM Tris base, 15 mM EDTA, 13% glycerol, 4% SDS, 2 mM PMSF), diluted 1:20 into RIPA buffer[60] containing 500 mM NaCl and 20 mM imidazole and incubated with pre-equilibrated Ni-NTA agarose at 4 °C for 1 h on a rotating platform. Subsequently, the Ni-NTA agarose was washed twice with RIPA buffer containing 500 mM NaCl and 30 mM imidazole, proteins were eluted with RIPA buffer containing 250 mM imidazole, and the Ni-NTA agarose beads were removed by centrifugation. The eluate was then diluted 1:5 into fresh RIPA buffer and subjected to immunoprecipitation with an anti-HA antiserum. The protein A-sepharose beads used in the

immunoprecipitation were subsequently washed twice, resuspended in buffer B (200 mM Tris pH 8, 15 mM EDTA, 13% glycerol, 4% SDS and 40 mM DTT) and heated at 95 °C for 5 min to release the bound proteins into the supernatant. Half of the supernatant was directly heated in SDS-PAGE sample buffer and the other half was diluted 1:20 into fresh RIPA buffer for another round of immunoprecipitation with an anti-FLAG antiserum. In experiments that involved the production of HA-UpaGΔ2(Y1735am/W1778A) and either $His_{10}$-UpaGΔ2(Y1735am) or $His_{10}$-UpaGΔ2(Y1735am/W1778A), proteins in both UV-irradiated and non-irradiated samples were precipitated with TCA. TCA pellets were then solubilized in buffer A and Ni-NTA purification was performed as described above. One-third of the eluted protein was resolved by SDS-PAGE, and the remainder was diluted 1:20 into fresh RIPA buffer and subjected to immunoprecipitation with an anti-HA antiserum prior to SDS-PAGE.

**Purification of a ~40 kD UpaGΔ2 crosslinking product.** An overnight culture of AD202 harboring pRS5 and pDULE-Bpa was washed and diluted into 1 l M9 at $OD_{600} = 0.05$. When the culture reached $OD_{600} = 0.5$, 1 mM IPTG and 1 mM Bpa were added and cells were incubated for an additional 1 h. The cells were collected by centrifugation (3000×*g*, 20 min, 4 °C) and the pellet was resuspended in 60 ml of the culture supernatant. Half of the cells were kept on ice and 10 ml aliquots of the other half were sequentially UV irradiated for 7 min in a 6-well-tissue culture plate. The irradiated samples were pooled and both untreated and irradiated cells were collected by centrifugation (3000×*g*, 20 min, 4 °C). The cell pellets were resuspended in a lysis buffer containing 50 mM Tris pH 8, 5 mM EDTA and 1× Halt Protease Inhibitor Cocktail (Thermo Fisher) and lysed using an Emulsiflex-C3 instrument. Unbroken cells were removed and cell membranes isolated by ultra-centrifugation (100,000×*g*, 45 min, 4 °C). The membranes were dispersed in PBS using a Dounce homogenizer and membrane-associated proteins were TCA precipitated. The TCA pellet was solubilized in buffer A and then diluted 1:20 in RIPA buffer containing 500 mM NaCl and 20 mM imidazole. The samples were incubated with pre-equilibrated Ni-NTA agarose at 4 °C for 1 h on a rotating platform. Subsequently, the Ni-NTA agarose was washed twice with RIPA buffer containing 500 mM NaCl and 30 mM imidazole. His-tagged UpaGΔ2 monomers and oligomers were eluted in RIPA buffer containing 500 mM NaCl, 250 mM imidazole and 1% glycerol, resolved by SDS-PAGE on 8–16% minigels, and visualized by Colloidal Blue staining. Gel slices containing proteins in the 35–40 kDa range were then analyzed by mass spectrometry.

**Mass spectrometry.** Gel slices containing proteins in the 35–40 kDa range were excised and minced finely in three independent experiments. The gel fragments were destained in a vial by agitating with 1.6 ml 50 mM $NH_4OAc$, 40% acetonitrile for 20 min (with one change of solution) and dehydrated by adding 1 ml acetonitrile (with one change of solution) for 10 min. Freshly heated and sonicated 1% sodium dodecanoate (160 µl) was then added and the samples were incubated at 90 °C for 20 min, sonicated for 2 h, and incubated with shaking at 40 °C for an additional 12 h. The supernatant was recovered and diluted to 1.6 ml with $H_2O$ and the pH was adjusted to 6–9. Trypsin (1 µg) was then added and the solution was incubated for ≥12 h at room temperature. Samples were acidified by adding 1% TFA and extracted twice with equal volumes of ethyl acetate to remove dodecanoic acid and placed at 50 °C under a stream of warm $N_2$. Additional peptides were obtained by adding 500 µl 0.4% formic acid, 40% acetonitrile to the original vials containing gel fragments and incubating for 15 min. This solution was combined with the dried down aqueous solution from the ethyl acetate extraction. After the samples were dried almost completely, 80 µl of 0.8% formic acid, 40% acetonitrile was added and they were agitated at 40 °C for 10 min. The samples were then diluted with 760 µl 0.4% formic acid and applied in series to C8 and C18 Empore StageTips[61]. Each tip/micro-column was washed twice with 100 µl 1.6% formic acid, 100 mM $NH_4OAc$, and once with 100 µl 1.6% formic acid. Reductive dimethylation was next performed essentially as described[62] to label the −UV and +UV samples with light and +4 per amino group chemistries, respectively. Columns were washed twice with 100 µl 1.6% formic acid, 100 mM $NH_4OAc$ before peptides were eluted with 100 µl 0.4% formic acid, 40% acetonitrile and 100 µl 0.4% formic acid, 80% acetonitrile. The 400 µl of material eluted from the two columns was mixed thoroughly before being applied to a single SCX StageTip/micro-column. This column was washed twice with 400 µl 0.4% formic acid, 80% acetonitrile and 400 µl 0.4% formic acid, 95% acetonitrile. Peptides were next eluted using 100 µl 5% $NH_4OH$, 40% acetonitrile, and 100 µl 5% $NH_4OH$, 80% acetonitrile. The eluate was dried down at 50 °C under $N_2$ and resuspended in 30 µl 0.2% TFA, 2% acetonitrile. A 10 µl portion of each sample was injected onto a trap configured Waters NanoAcquity column (180 µm × 20 mm Symmetry C18, 75 µm × 250mm BEH130 C18, held at 50 °C) using 0.1% formic acid as the "A" solvent and 0.1% formic acid in acetonitrile as the "B" solvent. Samples were trapped at a flow rate of 4 µl min$^{-1}$ for 12 min in 99.7% A. An optimized gradient was then run starting at 99.7% A at 30 s with a slight rise to 1% B by 16 min at a flow rate of 0.15 µl min$^{-1}$. Subsequently, the flow rate dropped to 0.1 µl min$^{-1}$ and a shallow gradient was formed raising B to 23% by 510 min followed by a steeper rise to 80% B by 599 min. One minute after switching out of trap configuration, a contact closure signal initiated data collection on a Q-Exactive Mass Spectrometer (Thermo Fisher). The instrument was set to collect a Top30 data set with MS1 collected at a resolution setting of 70 K, an AGC target of 3E6 and MS2 collected at a resolution setting 17.5 K and

an AGC target of 1E5 with an isolation window of 4 $M/Z$ and 100 ms maximum injection time. Only charge states 2–4 were considered for fragmentation with an undersell ratio of 3%. Data was collected for 600 min. Raw data was analyzed using MaxQuant 1.5.2.8[63] with standard settings, an *E. coli* protein database and a list of standard contaminants. A 1% false discovery rate was used for both peptides and proteins. Ratios used are uncorrected raw ratios.

**Trypsin and Lys-C digestion of the UpaGΔ2 crosslinked dimer**. AD202 harboring pRS4 and pDULE-Bpa were subjected to radiolabeling and UV irradiation as described above. Half of the cells were treated with PK and PMSF, and proteins in all samples were precipitated with TCA. Both PK treated and untreated samples were further divided in half and immunoprecipitations were performed using an anti-HA antiserum. Following the last wash step, the protein A-sepharose bead pellet and the adsorbed antibody-antigen complexes were resuspended in 10 μl RIPA buffer containing 100 ng μl⁻¹ trypsin and 20 mM CaCl₂, 100 ng μl⁻¹ Lys-C, or no protease and incubated for 1 h at 37 °C. Following protease treatment, NuPAGE LDS sample buffer (Thermo Fisher) containing 100 mM DTT was added to all samples. Proteins were then heated to 99 °C for 5 min and resolved by SDS-PAGE on Novex 12% Bis-Tris minigels in MES buffer (Thermo Fisher).

**Isolation of non-denatured His-tagged UpaGΔ2 oligomers**. AD202 harboring pRS5 and pDULE-Bpa were radiolabeled and subjected to UV irradiation. Cells collected at specific time points (10 ml aliquots) were collected by centrifugation and incubated in 1 ml M9 salts for 40 min on ice with 300 μg ml⁻¹ PK. PMSF (2 mM) was added to stop the protease digestion. Cells were then washed twice in cold M9 salts and lysed in 1 ml lysis buffer (50 mM Tris pH 8, 5 mM EDTA, 1× Halt protease inhibitor cocktail) by sonication (Misonix 3000 sonicator, Microtip). After unbroken cells were removed (3500×g, 10 min, 4 °C), cell membranes were isolated by ultracentrifugation (100,000×g, 45 min, 4 °C). The membranes were then resuspended in buffer C (50 mM NaH₂PO₄ pH 8, 500 mM NaCl) containing 20 mM imidazole, 1× Halt Protease Inhibitor Cocktail and 2% (w/v) *n*-Dodecyl-β-D-maltopyranoside (DDM) (Anatrace) and incubated on a rotating platform at 4 °C overnight. Insoluble material was removed by ultracentrifugation (100,000×g, 45 min, 4 °C) and the supernatant was incubated with pre-equilibrated Ni-NTA agarose at 4 °C on a rotating platform for 1 h. The Ni-NTA agarose was then washed twice with buffer C containing 30 mM imidazole and 0.05% DDM and His-tagged proteins were eluted in buffer C containing 250 mM imidazole and 0.05% DDM. The eluate was incubated at 25 °C or 99 °C in SDS sample buffer for 5 min prior to SDS-PAGE on 8–16% gels.

**Cell fractionation and urea extraction**. Cells were grown, subjected to pulse-chase radiolabeling, and UV irradiated as described above except that 10 ml of cells were harvested at each time point. Cells were collected by centrifugation (3000×g, 10 min, 4 °C), resuspended in 1 ml PBS and lysed by sonication. Unbroken cells were removed by centrifugation (3500×g, 10 min, 4 °C) and the supernatant was placed into a fresh tube. One-fifth of each sample was removed (total fraction) and the remainder was subjected to ultracentrifugation (100,000×g, 30 min, 4 °C) to pellet the membranes. After the supernatant (soluble fraction) was removed, the membrane pellet was resuspended in 20 mM Tris pH 7.4, 100 mM glycine, 6 M urea and incubated at 25 °C for 1 h. The membranes were then pelleted again as described above. The supernatant (urea-soluble fraction) was then removed and the pellet (urea-resistant fraction) was resuspended in PBS. TCA was added to all samples to precipitate proteins and immunoprecipitations were conducted using an anti-HA antiserum.

**Data availability**. All data that support the findings of this study are available from the corresponding author upon request.

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

## Acknowledgements

We would like to thank Matt Doyle and Sunyia Hussain for helpful comments on the manuscript. This work was supported by the Intramural Research Program of the National Institute of Diabetes and Digestive and Kidney Diseases.

## Author contributions

R.S. and H.D.B. designed the experiments. R.S., J.H.P., and D.E.A. performed the experiments. R.S., D.E.A., and H.D.B. analyzed the data. R.S. and H.D.B. wrote the paper.

## Additional information

**Competing interests:** The authors declare no competing financial interests.

