## [Peer Review File · Nature Communications]

Reviewers' comments:

Reviewer #1 (Remarks to the Author):

The Manuscript by Sikdar et al probes the assembly pathway of the trimeric autotransporter UpaG. Like other integral outer membrane proteins, it is assembled by the Bam complex. The recent crystal structures of the Bam complex revealed relatively weak interactions of the first and the last beta strands of BamA. When the strains are crosslinked with a disulfide bond, the Bam complex is not functional. This lead to the functional model, in which the BamA barrel domain open laterally towards the membrane where it accepts the substrate and promotes its folding. The manuscript by Sikdar et al, adds to the growing body of in vivo evidence, that beta barrel folding begins in the periplasm prior to the membrane insertion and hence argues against the folding at the BamA lateral gate. I think the paper makes an important contribution; the claims are well supported. I have some suggestions, which may help to strengthen several points.

1- When using Bpa crosslinking authors observed a formation of UpaG dimer. Authors argue that it is likely that the dimer is the product is an unstable trimer, in which two subunits are stabilized by crosslinking. Moreover, the unstable trimer is, in fact, an assembly intermediate. Although I agree that it's unlikely that a dimer is an intermediate for a stable trimer, I think the evidence for an unstable trimer intermediate is quite weak. The trimers on Fig. 4 are not nearly as impressive, possibly, because the trimer intermediate is unstable in SDS even without heating. Have authors considered using a general crosslinker to stabilize the trimer? The combination of a reversible general crosslinker (such as DSP or other) with Bpa can aid with evaluation if the Bpa-crosslinked dimer is a part of the unstable trimer. DSP can help to stabilize the trimer, but after BME treatment and DSP reversal, Bpa crosslinked dimer should remain stable. If crosslinking indeed stabilizes the trimer, it would be interesting to see if trimers can be detected in the periplasmic (or urea wash) faction.

2- In my opinion, the weakest point of the paper is the section about the interactions between two UpaG subunits/Bpa crosslinks (lines 222-248, Fig. 3C and D). First, the crosslinking radius of Bpa depends on the sequence context and can extend way beyond initially reported 3.1 A. For example, highly preferred amino acid methionine could be crosslinked as far as from the 16 A distance. Identification of Bpa crosslinked sites is extremely challenging and I find it naïve to pick the residues within 4 A on the structure (Fig. 3D). Also, it is impossible to know that 12 kDa peptides shown in Fig. 3C are indeed the products of the crosslinked dimer without mass spectrometry analysis. It may be simply a result of incomplete digest; no controls were performed to show that these peptides are present only in a UV crosslinked samples. There is no doubt that UpaG-Bpa crosslinked product is a dimer based on the work using differentially tagged UpaG (Fig. 3A). Therefore, I don't think this section contribute anything valuable to the paper, and I suggest removing it altogether. Authors may simply state that there are a number of residues of the second subunit, which lay within Bpa radius based on the calculated structure.

3- Authors identified an assembly-defective mutant W1778A. Because this region has been implicated in the substrate recognition by the Bam complex, authors argue that W1778A affects targeting to BamA. The reason behind the assembly defect is not that important for the scope of the paper, folding still begins in the periplasm, whether on the Bam or not – an open question. However, authors argue very strongly about the targeting defect of UpaG_W1778A and use it to conclude that the trimer intermediate forms a prior interaction with the Bam complex. There is no direct evidence to support it in the manuscript; the current data leave room for an alternative explanation. In my opinion, because UpaG_W1778A is not a substrate for a periplasmic protease DegP (data not shown – include in supplement), one can argue it is stalled on the Bam complex and hence isn't accessible. If W1778A stalls on the Bam complex, it can also explain the toxicity and is more consistent with the observation that slow growth promotes assembly. Slow growth is unlikely to improve recognition, but rather gives more time of substrate to fold on the Bam complex.

The fact that co-expression with the WT UpaG results in a mixed trimer formation isn't a good argument for targeting. It may very well be that W1778A affects a trimer stability, hence one or two copies of the WT UpaG (the stoichiometry is not resolved) can help to stabilize the trimer. The same point applies to the statement "a single functional β signal motif is sufficient to mediate the targeting reaction" (line 424). It was not determined if a single functional motif is sufficient. Unless authors provide further evidence that trimer formation occurs prior to the Bam recognition, the text should be modified to discuss multiple possibilities.

Line 95: "coiled-coli" should be coiled-coil

Reviewer #2 (Remarks to the Author):

Trimeric autotransporters are cell-surface-exposed, often virulence-related proteins of Gram-negative bacteria. They are anchored in the bacterial outer membrane via a 12-stranded β -barrel, to which each subunit contributes four β -strands. Using pulse-chase experiments combined with site-specific photocrosslinking, this manuscript provides novel insights into the biogenesis of such an oligomeric β -barrel.

Comments:

1. Evidence for the main conclusion of the paper, as reflected in the title ("Folding of a bacterial integral outer membrane protein is initiated in the periplasm"), was published already two decades ago by Eppens et al. ("Folding of a bacterial outer membrane protein during passage through the periplasm". 1997. EMBO J 16: 4295-4301). Nevertheless, I think this manuscript is very interesting. It has always remained enigmatic when and where oligomeric β -barrels are formed. The results described here demonstrate that the biogenesis of these proteins proceeds via a metastable trimer that is already formed in the periplasm, presumably before the protein interacts with the β -barrel assembly machinery (BAM). They also demonstrate that the C-terminal signal for recognition by the BAM has to be present in only one of the subunits. These are valuable new insights. Obviously, the work of Eppens et al. needs to be discussed, and the title and parts of the text need to be adopted accordingly and have to be focused on the oligomeric β -barrels.
2. Fig. S3 is very unclear. The Western blot (right panel) shows even more unspecific bands than the stained gel (left panel). How is this possible? Also, particularly in the right panel, the monomer and the dimer seem to be much more abundant than the trimer, in contrast to all other figures in the manuscript. Controls not expressing the UpaG construct should be included to help in the identification of the specific bands. In the left panel, two dashes indicate the position of the dimer. An error I presume?
3. The evidence for a metastable trimer in Fig. 4 is not totally convincing. The trimer appears as a double band. The upper one seems somewhat more intense in the unheated sample and the lower one in the heated sample. How should this be interpreted? Possibly, the use of a chemical crosslinker could fix more metastable trimers in the trimeric form and reinforce the data.
4. Lines 279-280: Indeed, OmpC appears to be in the membrane fraction and not extractable with urea in Fig 5, but at the earliest time point in Fig 7, it pretty much fractionates like the dimer of the UpaG construct in Fig 5. Possibly, the soluble and urea-extractable forms of OmpC in Fig 7 also represent assembly intermediates, which should be noted. Why is there such a difference between Figs 5 and 7 in this respect?
5. Lines 305-306: The effect of the mutation under these conditions cannot be evaluated, because Fig S8 does not show the results for the wild type.
6. Fig 6b: Is the band marked with an asterisk indeed the result of an unsuppressed amber codon? If so, why is this band much more intense in the W1778A mutant than in the wild type? Couldn't it be a degradation product?
7. As the W1778A mutation does apparently not inhibit the formation of the metastable trimer, I would expect to see crosslinked dimers of the two differently tagged versions of this mutant in the experiment shown in the lower panel of Fig 6C. Why were they not detected?

8. Lines 410-411: The suggestion that the linker segment might be in a hairpin conformation seems in conflict with the native-like conformation predicted based on the crosslinking data (Fig S6).

Minor issues:

1. Lines 47-50: This statement is not entirely correct. Although the mitochondrial outer membrane does contain some β -barrel proteins, the vast majority is α -helical.
2. Lines 151-152: It's unclear why the authors propose a role for the folded Him motif in protecting other parts of the protein against proteolysis. I guess each surface-exposed part of the protein should be folded in a protease-resistant conformation. β -Barrels are generally very resistant to proteases by themselves.
3. Line 207: Mass spec was not done with the HA construct but with the His10 construct.
4. Lines 253-254: I think the third subunit is already lost before sample analysis on the gels, that is during the TCA precipitations that are routinely done before Ni-NTA purification and/or IP according to Methods.
5. Line 547: Explain RIPA buffer.

Reviewer #3 (Remarks to the Author):

The manuscript by Bernstein and colleagues describes work on a member of the trimeric autotransporter adhesin (TAA) family and its Bam-assisted assembly/integration into the bacterial outer membrane (OM). A central question in the field is how the Bam machinery integrates client substrates into the OM; despite a relatively large amount of recent structural information it is not at all clear whether Bam sequentially handles single beta-strands or whether pre-folded beta-barrel intermediates are inserted as a unit (to name but two extreme possibilities for the Bam mechanism). The authors use a partial construct of the TAA UpaG, consisting of the C-terminal ~20 kD of the protein, including the C-terminal barrel domain and a small segment of the passenger. They then perform pulse-chase radiolabeling experiments to follow the integration of variously N-terminally tagged UpaG constructs into the OM. They also utilise proteinase K digestion to follow the appearance of stable products on the cell surface. Stable trimers appear with concomitant decrease of the monomer band, indicating appearance of the assembled TAA on the cell surface. PK digestion removes the N-terminal HA tag, so that the trimer band is not detected with HA-immunoprecipitation (Fig. 1). In Fig 2, a photo-crosslinker (Bpa) is introduced at the site of a conserved, exposed tyrosine in the UpaG barrel. Analysis with and without UV-irradiation shows the appearance of a dimeric species that is PK protected (Fig. 2), suggesting it is a periplasmic intermediate. Fig 3 presents data to demonstrate this species is a UpaG dimer and attempts to assign its identity, utilising two proteases that cleave after basic residues. The authors then show data to support their assertion that the dimer is in fact an unstable trimer (Fig. 4). Fractionations and urea treatments demonstrate that the intermediate is present in all fractions (soluble and membrane), indicating several distinct intermediate species (Fig. 5). Finally, utilising a C-terminal Bam targeting mutant it is shown that no stable trimer is integrated into the membrane, unless at least one of the monomers in the trimer is native (Figs 6 and 7). A model for proposed Bam action is then presented in which a partially folded, barrel-like intermediate is integrated by the Bam complex (Fig. 8).

1. Lines 34-35 of the abstract are unclear ("persists until the termination of passenger domain translocation").
2. Line 46: why do you use "appear"? Isn't this established by now?
3. Lines 110-112: unclear ("The data....OM proteins").
4. Line 159: it is hard to say whether the monomeric form is really resistant towards degradation because it also disappears due to the assembly process.
5. Line 191: please show data in supplementary.

6. Line 234: the section about the assignment of the dimeric species is unclear to me. We need more information here. What is the assignment based on? At face value, the size of the band is more like 15-16 kDa rather than the stated 11-12 kD. This has also consequences for the section of lines 245-248 and 265-267: we need to be certain about the assignment of the fragment before we can conclude that the intermediate has native-like structure.

7. Line 357-358: given that both constructs are expressed from one promoter, the question is whether they are expressed at the same level. Are the same results obtained when the WT and mutant variants are swapped on the expression plasmid?

8. Line 385: the levels of the dimer intermediate with the C-terminal targeting mutant are much lower than for the WT. Does this mean that folding in the periplasm depends on an interaction with Bam?

9. Line 409-410: the statement that the linker segments are embedded native-like inside the barrel is very speculative. What really is the evidence supporting this?

10. Line 413: the reference to data in a manuscript in preparation should be removed. The same goes for line 430.

11. Line 434: with Bpa crosslinkers at various positions (which ones?) no interactions with Bam components were observed. Isn't that strange? Do you always see the intermediate dimer species? If so, that would argue against a specific intermediate. What about interactions with periplasmic chaperones?

The term "translocation" should be defined a bit better. I assume this refers to the transmembrane movement of the passenger domain?

12. Line 444: I would not say that the evidence for trimerisation is strong. Evidence for the dimer is convincing, that for the trimer is more circumstantial.

Figure 3: panel d is not very clear and needs labels. Lines have been sized wrongly. Also, what are the blue bits?

Figure 4: the difference in trimer band intensity between both temperatures is not very large. Can you do densitometry to quantify? Why would there be trimer bands after boiling if as stated the stability of the species is low?

It also seems to me that the increase in dimer intensity upon boiling is too much to be accounted for by the decrease of the trimer band?

Why would the HA antibody stain the different oligomeric species in a different way? Can you show that in a control experiment? (ie load identical amounts of stable trimer and urea unfolded monomer)

Figure 8: I am not sure if the evidence in this manuscript fully supports the fairly detailed folding and assembly model presented in this figure. Less well-supported steps are the existence of an asymmetric trimer, the presence of native-like structure in this species, and step 3.

RESPONSE TO REVIEWERS' COMMENTS

(Please note that the line numbers have changed in the revised manuscript)

Reviewer #1 (Remarks to the Author):

...The manuscript by Sikdar et al, adds to the growing body of in vivo evidence, that beta barrel folding begins in the periplasm prior to the membrane insertion and hence argues against the folding at the BamA lateral gate. I think the paper makes an important contribution; the claims are well supported. I have some suggestions, which may help to strengthen several points.

1- When using Bpa crosslinking authors observed a formation of UpaG dimer. Authors argue that it is likely that the dimer is the product is an unstable trimer, in which two subunits are stabilized by crosslinking. Moreover, the unstable trimer is, in fact, an assembly intermediate. Although I agree that it's unlikely that a dimer is an intermediate for a stable trimer, I think the evidence for an unstable trimer intermediate is quite weak. The trimers on Fig. 4 are not nearly as impressive, possibly, because the trimer intermediate is unstable in SDS even without heating. Have authors considered using a general crosslinker to stabilize the trimer? The combination of a reversible general crosslinker (such as DSP or other) with Bpa can aid with evaluation if the Bpa-crosslinked dimer is a part of the unstable trimer. DSP can help to stabilize the trimer, but after BME treatment and DSP reversal, Bpa crosslinked dimer should remain stable. If crosslinking indeed stabilizes the trimer, it would be interesting to see if trimers can be detected in the periplasmic (or urea wash) fraction.

We would like to emphasize that the results presented in Fig. 4 are not the only evidence for the existence of an unstable trimer. As we point out (lines 362-366), the co-fractionation of the monomeric form of UpaG Δ 2 with the crosslinked dimer also suggests that the dimer is derived from a trimeric intermediate. In addition, the presence of the crosslinked dimer in the urea-resistant (membrane-embedded) fraction is difficult to explain unless the dimer is part of a trimeric barrel-like structure that has an entirely hydrophobic outer surface. We did attempt to stabilize the trimer by chemical crosslinking before we submitted the manuscript, but to address the reviewers' concern we have repeated the experiment several times using a wider range of chemical crosslinkers (including DMP, DSP, DSS, DSG, TSAT and EGS) and incubation times (20-45 minutes on ice). Unfortunately, the chemical crosslinking experiments did not yield interpretable results. The dimer disappeared after the addition of some of the crosslinkers (which indicates that a chemical reaction occurred), but only a heterogeneous mixture of high molecular weight crosslinking products were observed. We now indicate in the text (line 320) that chemical crosslinking did not produce useful insights.

2- In my opinion, the weakest point of the paper is the section about the interactions between two UpaG subunits/Bpa crosslinks (lines 222-248, Fig. 3C and D). First, the crosslinking radius of Bpa depends on the sequence context and can extend way beyond initially reported 3.1 Å. For example, highly preferred amino acid methionine could be crosslinked as far as from the 16 Å distance. Identification of Bpa crosslinked sites is extremely challenging and I find it naïve to pick the residues within 4 Å on the structure (Fig. 3D). Also, it is impossible to know that 12 kDa peptides shown in Fig. 3C are indeed the products of the crosslinked dimer without mass spectrometry analysis. It may be simply a result of incomplete digest; no controls were performed to show that these peptides are present only in a UV crosslinked samples. There is no doubt that UpaG-Bpa crosslinked product is a dimer based on the work using differentially tagged UpaG (Fig. 3A). Therefore, I don't think this section contribute anything valuable to the paper, and I suggest removing it altogether. Authors may simply state that there are a number

of residues of the second subunit, which lay within Bpa radius based on the calculated structure.

While the reviewer is correct that crosslinks > 4Å away from Bpa have been reported, the crosslinking is presumably due to an unanticipated interaction that brings together the photoactivatable amino acid and the target residue. After all, the amino acid analog is always physically constrained by the polypeptide backbone and cannot extend more than 4Å away. In the paper that reports the methionine “magnet” phenomenon [Wittelsberger, A. et al., FEBS Letters (2006) 580: 1872], the authors suggest that the crosslinking to distant residues is due to an unexpectedly high degree of conformational flexibility in a GPCR that is not apparent in its crystal structure. In any case, we believe that Figs 3C and 3D add significant information to the paper. First, to address one of reviewer’s main concerns, we have now added a control experiment showing that the 12 kD fragments are only present in UV-irradiated samples (new Supplementary Fig. 6) and modified the text accordingly (line 290). To address another concern (as well as a concern raised by reviewer 3), we have also added a possible explanation for the difference between the observed size of the large tryptic/Lys-C fragments in Fig. 3C (slightly > 12 kD) and the expected size (11-12 kD). As we suggest (lines 300-310), the formation of a covalent bond between Bpa and one of the residues that are within 4Å in the final folded structure (Fig. 3D) might create steric hindrance that blocks trypsin/Lys-C cleavage at the upstream lysine residue and leads to incomplete digestion. While this observation does not enable us to pinpoint the exact residue in the second subunit that forms a bond with Bpa, we believe that it provides interesting circumstantial evidence that should be presented. By the same token, we have tried to word the text as carefully as possible to indicate that Figs. 3C and 3D only provide hints about the conformation of the UpaG Δ 2 assembly intermediate represented by the crosslinked dimer (lines 310-313).

3- Authors identified an assembly-defective mutant W1778A. Because this region has been implicated in the substrate recognition by the Bam complex, authors argue that W1778A affects targeting to BamA. The reason behind the assembly defect is not that important for the scope of the paper, folding still begins in the periplasm, whether on the Bam or not – an open question. However, authors argue very strongly about the targeting defect of UpaG_W1778A and use it to conclude that the trimer intermediate forms a prior interaction with the Bam complex. There is no direct evidence to support it in the manuscript; the current data leave room for an alternative explanation. In my opinion, because UpaG_W1778A is not a substrate for a periplasmic protease DegP (data not shown – include in supplement), one can argue it is stalled on the Bam complex and hence isn’t accessible. If W1778A stalls on the Bam complex, it can also explain the toxicity and is more consistent with the observation that slow growth promotes assembly. Slow growth is unlikely to improve recognition, but rather gives more time of substrate to fold on the Bam complex. The fact that co-expression with the WT UpaG results in a mixed trimer formation isn’t a good argument for targeting. It may very well be that W1778A affects a trimer stability, hence one or two copies of the WT UpaG (the stoichiometry is not resolved) can help to stabilize the trimer. The same point applies to the statement “a single functional β signal motif is sufficient to mediate the targeting reaction” (line 424). It was not determined if a single functional motif is sufficient. Unless authors provide further evidence that trimer formation occurs prior to the Bam recognition, the text should be modified to discuss multiple possibilities.

The reviewer appears to have misunderstood the fate of the UpaG Δ 2 (W1778A) mutant. As we show in multiple Figures and state in the text (e.g., lines 386-388 and 416-418), a large fraction of the mutant protein is degraded. The results do not suggest that the mutant is stalled on the Bam complex in an “inaccessible” form. For reasons that we do not understand, the mutant protein is degraded in a strain that lacks DegP. This observation suggests that the mutant is

degraded by another protease, like a subset of other unstable outer membrane proteins [see, for example, Echenique-Rivera, H. et al. (2011) *Infect Immun* 79: 4308; Narita, S. et al. (2013) *Proc Natl Acad Sci USA* 110: E3612; Weski, J. and Ehrmann, M. (2012) *J Bacteriol* 194: 3225]. In any case, because the identity of the protease that degrades the UpaG Δ 2 mutant is peripheral to our study we have removed the reference to DegP from the manuscript. Furthermore, we would also like to note that the presence of a large fraction of the crosslinked UpaG Δ 2 (W1778A) dimer in the soluble fraction provides evidence that this assembly intermediate does not form only after the protein interacts with the Bam complex.

To clarify the toxicity of the UpaG Δ 2 mutant, we have also rephrased (lines 398-399) As it turns out, the toxicity was only observed in the crosslinking experiments described later in the paragraph and seems to be associated with the presence of two plasmids. In writing this passage we have tried to walk a fine line between providing an explanation for the use of a different expression system and providing a lot of minute technical details that might confuse readers. In any case, the results do not seem to suggest that the mutant protein stalls on the Bam complex.

It seems very unlikely to us that the presence of a wild-type copy of UpaG Δ 2 in a mixed trimer facilitates assembly of the mutant protein simply by stabilizing it. If that were the case, we would have expected to see that the residual mutant protein (i.e., the protein that is not degraded by periplasmic proteases) would have formed stable trimers without the aid of the wild-type protein. The dimers formed by the UpaG Δ 2 (W1778A) mutant are neither degraded nor assembled into a stable trimer and, from our fractionation studies, are clearly not integrated into the OM. The results that we obtained strongly suggest that the expression of the wild-type protein produces a qualitative rather than just a quantitative effect on the assembly of the mutant protein. In addition, we do not mean to imply that we have determined the stoichiometry of the wild-type and mutant protein subunits in the mixed complexes. Nevertheless, to address the reviewer's concern we have modified the text and now state that the results only "suggest that a single functional β signal motif may be sufficient to mediate the targeting reaction" (line 554) and have removed "[targeted] by a single β signal motif" from the legend to Fig. 8.

Line 95: "coiled-coli" should be coiled-coil

We have corrected this error.

Reviewer #2 (Remarks to the Author):

Trimeric autotransporters are cell-surface-exposed, often virulence-related proteins of Gram-negative bacteria. They are anchored in the bacterial outer membrane via a 12-stranded β -barrel, to which each subunit contributes four β -strands. Using pulse-chase experiments combined with site-specific photocrosslinking, this manuscript provides novel insights into the biogenesis of such an oligomeric β -barrel.

Comments:

1. Evidence for the main conclusion of the paper, as reflected in the title ("Folding of a bacterial integral outer membrane protein is initiated in the periplasm"), was published already two decades ago by Eppens et al. ("Folding of a bacterial outer membrane protein during passage through the periplasm". 1997. *EMBO J* 16: 4295-4301). Nevertheless, I think this manuscript is very interesting. It has always remained enigmatic when and where oligomeric β -barrels are

formed. The results described here demonstrate that the biogenesis of these proteins proceeds via a metastable trimer that is already formed in the periplasm, presumably before the protein interacts with the β -barrel assembly machinery (BAM). They also demonstrate that the C-terminal signal for recognition by the BAM has to be present in only one of the subunits. These are valuable new insights. Obviously, the work of Eppens et al. needs to be discussed, and the title and parts of the text need to be adopted accordingly and have to be focused on the oligomeric β -barrels.

We agree with the reviewer that the paper by Eppens et al. is highly relevant to our work and we now cite it in the Discussion (p. 19, ref. 48). Despite its title, however, this paper does not provide direct evidence that outer membrane proteins fold in the periplasm. When the paper was written 20 years ago it was not yet clear that β barrel proteins pass through the periplasm on their way to the outer membrane. To address this question, the authors introduced two cysteine residues into PhoE at residues in a loop and in the barrel wall that are within hydrogen bonding distance in the final structure. They then showed that a disulfide bond forms between the two cysteine residues in the periplasm and that the modified protein can insert into the outer membrane. While the experiments provided elegant proof that outer membrane proteins pass through the periplasm, they did not show where native outer membrane proteins normally fold. Instead, they showed that the formation of very limited tertiary structure is tolerated by the outer membrane protein assembly machinery (the identity of which was unknown at the time). While the results are consistent with our main conclusion, the paper really addresses a different issue.

2. Fig. S3 is very unclear. The Western blot (right panel) shows even more unspecific bands than the stained gel (left panel). How is this possible? Also, particularly in the right panel, the monomer and the dimer seem to be much more abundant than the trimer, in contrast to all other figures in the manuscript. Controls not expressing the UpaG construct should be included to help in the identification of the specific bands. In the left panel, two dashes indicate the position of the dimer. An error I presume?

This is the only Figure in the paper (Supplementary Fig. 4 in the revised manuscript) that shows a Colloidal Blue stained gel and a Western blot; all of the other Figures show immunoprecipitations. Because the method of detection was different, it is not surprising that the signal intensities are also different. Western blots are much more sensitive than stained gels, so additional background bands are typically observed. Likewise, the stronger monomer and dimer signals observed on the Western blot may be due to differences in transfer efficiency or other technical quirks of the Western blotting method. Most of the bands observed on the stained gel are also observed on the Western blot and therefore are probably UpaG derivatives. For this reason it is unlikely that controls that do not express the UpaG Δ 2 construct would be useful. Finally, we have corrected the error mentioned by the reviewer.

3. The evidence for a metastable trimer in Fig. 4 is not totally convincing. The trimer appears as a double band. The upper one seems somewhat more intense in the unheated sample and the lower one in the heated sample. How should this be interpreted? Possibly, the use of a chemical crosslinker could fix more metastable trimers in the trimeric form and reinforce the data.

Based on its mobility, the upper band very likely corresponds to the trimer and has been labeled accordingly. This issue is further addressed in our reply to reviewer 3 (comment 12). As noted in our reply to reviewer 1 (see above), chemical crosslinking experiments have not yielded interpretable results.

4. Lines 279-280: Indeed, OmpC appears to be in the membrane fraction and not extractable with urea in Fig 5, but at the earliest time point in Fig 7, it pretty much fractionates like the dimer of the UpaG construct in Fig 5. Possibly, the soluble and urea-extractable forms of OmpC in Fig 7 also represent assembly intermediates, which should be noted. Why is there such a difference between Figs 5 and 7 in this respect?

We agree with the reviewer that the soluble and urea-extractable forms of OmpC observed at the first time point in Fig. 7 might be assembly intermediates. The slight difference in the apparent behavior of OmpC in Figs 5 and 7, however, is probably due at least in part to a difference in the exposure of the two gels. In any case, because we have not studied the assembly of OmpC very carefully at this point, we would rather not speculate on the significance of the soluble and urea-extractable forms.

5. Lines 305-306: The effect of the mutation under these conditions cannot be evaluated, because Fig S8 does not show the results for the wild type.

To address this concern we have removed the comparison to wild-type UpaG Δ 2.

6. Fig 6b: Is the band marked with an asterisk indeed the result of an unsuppressed amber codon? If so, why is this band much more intense in the W1778A mutant than in the wild type? Couldn't it be a degradation product?

We agree that a degradation product of the mutant protein might co-migrate with the amber fragment in Fig. 6B. To address the reviewer's concern we have now changed the asterisk to a double asterisk and noted in the Figure legend that the band may contain a degradation product (lines 1053-1054).

7. As the W1778A mutation does apparently not inhibit the formation of the metastable trimer, I would expect to see crosslinked dimers of the two differently tagged versions of this mutant in the experiment shown in the lower panel of Fig 6C. Why were they not detected?

A small amount of the crosslinked dimer was indeed detected and perhaps can be seen best by enlarging the Figure. Because we detect much less of the mutant dimer than the wild-type dimer (see Fig. 6B), we would expect the signal produced by the heterodimer in Fig. 6C to be much lower in the bottom gel than in the top gel. In Fig. 6C we used an exposure of the two gels that we believe shows the results clearly even though the signal produced by the heterodimer in the bottom gel is weak.

8. Lines 410-411: The suggestion that the linker segment might be in a hairpin conformation seems in conflict with the native-like conformation predicted based on the crosslinking data (Fig S6).

The reviewer raises a good point here. We have tried to clarify the model by suggesting that a hairpin forms between the linker and the passenger domain rather than within the linker itself (lines 524-526). This idea is consistent with the results of a previous study (ref. 28) that suggest that the linker contains a flexible region located near the extracellular side of the β barrel. As an aside, available evidence indicates that the linker segment of classical (monomeric) autotransporters is in a native-like conformation at a relatively early stage of assembly before passenger domain translocation is complete (ref. 45).

Minor issues:

1. Lines 47-50: This statement is not entirely correct. Although the mitochondrial outer membrane does contain some β -barrel proteins, the vast majority is α -helical.

To address this concern we have removed the statement that the outer membranes of organelles of bacterial origin contain predominantly β barrel proteins.

2. Lines 151-152: It's unclear why the authors propose a role for the folded Him motif in protecting other parts of the protein against proteolysis. I guess each surface-exposed part of the protein should be folded in a protease-resistant conformation. β -Barrels are generally very resistant to proteases by themselves.

The reviewer raises a good point here. We do not mean to imply that the Him motif protects other parts of the protein against proteolysis. We have rewritten the sentence to improve clarity.

3. Line 207: Mass spec was not done with the HA construct but with the His10 construct.

We have corrected this error.

4. Lines 253-254: I think the third subunit is already lost before sample analysis on the gels, that is during the TCA precipitations that are routinely done before Ni-NTA purification and/or IP according to Methods.

The reviewer makes a good point, but we did not perform TCA precipitations in all of our experiments. To cover all the bases we now state that "the third subunit would dissociate from the complex *during sample preparation* or upon heating in SDS".

5. Line 547: Explain RIPA buffer.

We have now provided a reference for this buffer.

Reviewer #3 (Remarks to the Author):

The manuscript by Bernstein and colleagues describes work on a member of the trimeric autotransporter adhesin (TAA) family and its Bam-assisted assembly/integration into the bacterial outer membrane (OM). A central question in the field is how the Bam machinery integrates client substrates into the OM; despite a relatively large amount of recent structural information it is not at all clear whether Bam sequentially handles single beta-strands or whether pre-folded beta-barrel intermediates are inserted as a unit (to name but two extreme possibilities for the Bam mechanism). The authors use a partial construct of the TAA UpaG, consisting of the C-terminal ~20 kD of the protein, including the C-terminal barrel domain and a small segment of the passenger [to address this question]...

1. Lines 34-35 of the abstract are unclear ("persists until the termination of passenger domain translocation").

The reviewer does not explain how the clarity of this phrase might be improved. It would not be possible to provide more detailed information because the Abstract is limited to 150 words.

2. Line 46: why do you use "appear"? Isn't this established by now?

We use "appear" to be cautious. Although available evidence indicates that the transmembrane segments of multispanning α -helical membrane proteins partition into the lipid bilayer in a stepwise fashion, the assembly of these proteins has not been studied in detail. We believe that if we are not cautious some investigators may feel that we have overstated the evidence.

3. Lines 110-112: unclear ("The data....OM proteins").

The reviewer does not indicate how the clarity of this sentence might be improved.

4. Line 159: it is hard to say whether the monomeric form is really resistant towards degradation because it also disappears due to the assembly process.

The disappearance of the monomer over time due to the assembly process is not relevant. Our conclusion is based on the observation that the level of the monomer detected at each time point did not significantly change after the addition of proteinase K. The same result was obtained in multiple experiments (see Figs. 1, 2, 6, and Supplementary Figs. 2 and 10). In Fig. 1, for example, one should compare lanes 1 and 6, 2 and 7, 3 and 8, etc.

5. Line 191: please show data in supplementary.

As suggested by the reviewer, we now show the kinetics of assembly of the His- and FLAG-tagged versions of UpaG Δ 2 in Supplementary Fig. 3.

6. Line 234: the section about the assignment of the dimeric species is unclear to me. We need more information here. What is the assignment based on? At face value, the size of the band is more like 15-16 kDa rather than the stated 11-12 kD. This has also consequences for the section of lines 245-248 and 265-267: we need to be certain about the assignment of the fragment before we can conclude that the intermediate has native-like structure.

Please see the reply to reviewer 1 (comment 2). In brief, we have modified the text to provide a possible explanation for the apparent difference between the observed and predicted fragment sizes (lines 300-310). As we now indicate, the residues highlighted in Fig. 3D are very close to the upstream lysine residue, and crosslinking between Bpa and these residues might create a steric block that would lead to incomplete protease digestion and the formation of larger-than-expected protease fragments.

7. Line 357-358: given that both constructs are expressed from one promoter, the question is whether they are expressed at the same level. Are the same results obtained when the WT and mutant variants are swapped on the expression plasmid?

There is no reason to suspect a priori that the two genes (both of which have identical Shine-Dalgarno sequences) are expressed at significantly different levels. In any case, the exact level of expression is not important here because the experiment shown in Fig. 6C is qualitative. Our goal is simply to show that mixed dimers and trimers are formed. A change in the amounts of mixed dimers and trimers that are formed due to differences in expression levels would not change the interpretation of the results.

8. Line 385: the levels of the dimer intermediate with the C-terminal targeting mutant are much

lower than for the WT. Does this mean that folding in the periplasm depends on an interaction with Bam?

We observed a much lower amount of the mutant dimer most likely because the mutant protein is relatively unstable. Presumably a significant fraction of the protein is recognized by yet unidentified periplasmic proteases and degraded because it cannot interact with the Bam complex.

9. Line 409-410: the statement that the linker segments are embedded native-like inside the barrel is very speculative. What really is the evidence supporting this?

As part of our model, we propose that the linker segments might be embedded inside a barrel-like structure at this stage, but we do not say anything about a “native-like” conformation. While we agree that the statement is somewhat speculative, we describe published and unpublished evidence in the next sentence (lines 527-528) that are consistent with our model.

10. Line 413: the reference to data in a manuscript in preparation should be removed. The same goes for line 430.

We have removed the first reference but we believe that the second reference to unpublished results provides useful information that helps to support our model. We will certainly remove this reference, however, if the citation of unpublished results is incompatible with journal policy (we could not find a discussion of this topic in the Instructions to Authors).

11. Line 434: with Bpa crosslinkers at various positions (which ones?) no interactions with Bam components were observed. Isn't that strange? Do you always see the intermediate dimer species? If so, that would argue against a specific intermediate. What about interactions with periplasmic chaperones?

In this sentence we are simply trying to explain to the reader why we were unable to take our analysis one step further, but the lack of crosslinking to the Bam complex in no way affects any of our conclusions. Negative results are not unusual in site-specific crosslinking experiments because the formation of a crosslink is highly position-specific, and sometimes even well-established protein-protein interactions are difficult to detect using this technique. By the same token, positive results are generally highly reproducible and can provide significant insights. We have observed very efficient dimer formation in every crosslinking experiment that we have performed with the UpaGΔ2(Y1735am) construct (at least 30 experiments so far).

The term "translocation" should be defined a bit better. I assume this refers to the transmembrane movement of the passenger domain?

“Translocation” is a standard term that has been used for more than 40 years to describe the movement of a polypeptide chain across a biological membrane. We believe that readers will be familiar with this term.

12. Line 444: I would not say that the evidence for trimerisation is strong. Evidence for the dimer is convincing, that for the trimer is more circumstantial.

To address this concern we have removed the word “strong”.

Figure 3: panel d is not very clear and needs labels. Lines have been sized wrongly. Also, what are the blue bits?

The blue bits and the misplaced line are errors that were inadvertently introduced during image processing and resizing. We have now corrected these errors.

Figure 4: the difference in trimer band intensity between both temperatures is not very large. Can you do densitometry to quantify? Why would there be trimer bands after boiling if as stated the stability of the species is low?

To examine assembly intermediates in this experiment, we first needed to treat cells with proteinase K to remove surface exposed passenger domains that were associated with fully assembled trimers. Although we repeated the experiment several times, a very small fraction of the surface exposed passenger domain remained resistant to proteinase K digestion. Due to this technical glitch, a small amount of trimer is always observed even after the samples are boiled. The small fraction of protease-resistant stable trimers creates an unavoidable background signal that hinders quantitation of the results.

It also seems to me that the increase in dimer intensity upon boiling is too much to be accounted for by the decrease of the trimer band?

For reasons that are explained below, it is not possible to make quantitative comparisons of the different oligomeric forms of UpaG Δ 2.

Why would the HA antibody stain the different oligomeric species in a different way? Can you show that in a control experiment? (ie load identical amounts of stable trimer and urea unfolded monomer)

It should be noted that all of the experiments (except for the experiment shown in Supplementary Fig. 4) are immunoprecipitations, not Western blots. It seems likely that the anti-HA antibody immunoprecipitated the oligomeric forms of UpaG Δ 2 more efficiently than the monomeric form at least in part because the oligomers contain multiple HA tags instead of a single HA tag. Consistent with this explanation, the anti-UpaG antiserum, which presumably recognizes multiple epitopes, appeared to immunoprecipitate each isoform of the protein with similar efficiency. In principle, the efficiency with which each isoform of UpaG Δ 2 is immunoprecipitated might also be influenced by other factors such as the accessibility of the relevant epitopes. We attempted to perform the control experiment suggested by the reviewer, but the experiment was confounded by our inability to fully denature the stable trimer even after boiling in 8M urea.

Figure 8: I am not sure if the evidence in this manuscript fully supports the fairly detailed folding and assembly model presented in this figure. Less well-supported steps are the existence of an asymmetric trimer, the presence of native-like structure in this species, and step 3.

As we state in the text, the illustration shown in Fig. 8 is only a model. Although some steps may be better supported than others, we believe that our model is a reasonable proposal based on the results presented in the manuscript. To address the reviewer's concern (and a concern of reviewer 2), we have now toned down our argument for the existence of an asymmetric trimer in the Results, Discussion and legend to Figure 8 by stating that our results "suggest" (rather than "strongly suggest") the existence of such an intermediate. If one is willing to accept the proposal

that the crosslinked dimer is derived from an asymmetrical trimer, then the existence of a urea-resistant population of crosslinked dimers strongly supports step 3.

REVIEWERS' COMMENTS:

Reviewer #1 (Remarks to the Author):

Related to the comment 2:

The nature of crosslinking peptides cannot be determined without a mass spectrometry analysis. Crosslinked peptides often migrate outside of the calculate molecular weight. Indeed, crosslinking on more distant sites can be "due to an unanticipated interaction that brings together the photoactivatable amino acid and the target residue". In addition, there is no UpaG structure, the residues/peptides are depicted using a structure of the homologous protein. You can keep it if you insist, but this evidence is indeed very circumstantial.

Related to the comment 3:

Assembly-defective substrates arrested on the Bam complex can be still degraded on the Bam complex by an associated protease, for example, BepA, while being inaccessible to periplasmic proteases such as DegP. While fractionation clearly shows that UpaG Δ 2 (W1778A) isn't membrane integrated, it does not prove that the dimer/trimer forms prior the interaction with the Bam complex. It is possible that UpaG Δ 2 (W1778A) is loosely associated with a periplasmic ring of the Bam complex etc.

Since the toxicity observed in some experiments is caused by two-plasmid system and not an expression of the mutant protein, I strongly encourage deleting the statement line 398-399 because it is very misleading.

Reviewer #2 (Remarks to the Author):

The authors' responses to the issues raised in my previous report are acceptable. While reading the revised manuscript, I spotted a few minor errors:

1. Lines 278 and 288: Fig S5A shows that the Lys-C fragment is four amino-acid residues larger than the tryptic fragment, i.e., it should contain 54 instead of 53 residues.
2. Fig S5B: The first blue asterisk, marking position 15 in the alignment, is a mistake. Trypsin will cleave after the lysine that is present in this position in UpaG; hence, this lysine is not part of the large tryptic fragment.
3. Fig 3B: I don't think the number of decimals shown in the last column is in accordance with the accuracy of the method used.

Reviewer #3 (Remarks to the Author):

In their revised manuscript the authors have addressed most of my questions and concerns. A few small issues remain:

1. Abstract: change "until the termination of passenger domain translocation" to "until the termination of passenger domain translocation through the UpaG barrel". I assume a few words over the length limit for the abstract will be OK.
2. Line 52: change "appear to partition" into "likely partition". I don't think many people would get upset about this.

3. Finally, regarding point 10, I maintain that the reference to unpublished data should be removed regardless of journal policy. How can "unpublished results provide useful information"?

REPLY TO REVIEWERS' COMMENTS

Reviewer #1

Related to the comment 2:

The nature of crosslinking peptides cannot be determined without a mass spectrometry analysis. Crosslinked peptides often migrate outside of the calculate molecular weight. Indeed, crosslinking on more distant sites can be “due to an unanticipated interaction that brings together the photoactivatable amino acid and the target residue”. In addition, there is no UpaG structure, the residues/peptides are depicted using a structure of the homologous protein. You can keep it if you insist, but this evidence is indeed very circumstantial.

While it is certainly true that we do not know the structure of the assembly intermediate represented by the crosslinked dimer, we believe that we would be remiss in not taking advantage of the available crystal structures to illustrate an intriguing possibility that emerges from the peptide mapping data. We have been careful to indicate that Fig. 3D shows only one possibility.

Related to the comment 3:

Assembly-defective substrates arrested on the Bam complex can be still degraded on the Bam complex by an associated protease, for example, BepA, while being inaccessible to periplasmic proteases such as DegP. While fractionation clearly shows that UpaG Δ 2 (W1778A) isn't membrane integrated, it does not prove that the dimer/trimer forms prior the interaction with the Bam complex. It is possible that UpaG Δ 2 (W1778A) is loosely associated with a periplasmic ring of the Bam complex etc.

It is very unlikely that the mutant interacts with the Bam complex before assembly because a large amount of the crosslinked dimer was found in the soluble (periplasmic) fraction. To argue that the mutant protein interacts with the Bam complex, it would be necessary to invoke a model in which the monomer binds to the Bam complex and then dissociates at some point after assembly has been initiated. Although such a “hit and run” model seems difficult to imagine, it cannot be completely excluded. For this reason we have tried to be very cautious by stating at several places that the mutant protein fails to interact “productively” with the Bam complex (see lines 342 and 410) and that our results “strongly suggest that the mutant protein begins to fold but effectively remains trapped in the periplasm” (lines 354-355).

Since the toxicity observed in some experiments is caused by two-plasmid system and not an expression of the mutant protein, I strongly encourage deleting the statement line 398-399 because it is very misleading.

We cannot completely remove lines 398-399 because they provide a rationale for switching expression systems, but to address the reviewer's concern we have revised this statement to indicate explicitly that the toxicity was seen when we transformed cells with two plasmids.

Reviewer #2:

The authors' responses to the issues raised in my previous report are acceptable. While reading the revised manuscript, I spotted a few minor errors:

1. Lines 278 and 288: Fig S5A shows that the Lys-C fragment is four amino-acid residues larger than the tryptic fragment, i.e., it should contain 54 instead of 53 residues.

We have corrected this error.

2. Fig S5B: The first blue asterisk, marking position 15 in the alignment, is a mistake. Trypsin will cleave after the lysine that is present in this position in UpaG; hence, this lysine is not part of the large tryptic fragment.

We have corrected the Figure Legend to indicate that we have also placed an asterisk above the lysine residue located N-terminal to the large tryptic fragment. As we note, the formation of a crosslink to the lysine residue would likely block cleavage by trypsin. Indeed trypsin treatment of the crosslinked dimer may have yielded an unexpectedly large product (Fig. 3C) because this lysine was inaccessible to the protease.

3. Fig 3B: I don't think the number of decimals shown in the last column is in accordance with the accuracy of the method used.

To address this concern we have removed the last two decimal places.

Reviewer #3:

In their revised manuscript the authors have addressed most of my questions and concerns. A few small issues remain:

1. Abstract: change "until the termination of passenger domain translocation" to "until the termination of passenger domain translocation through the UpaG barrel". I assume a few words over the length limit for the abstract will be OK.

We believe that the suggested change would be misleading. It is currently unclear if the passenger domain is secreted through a channel formed exclusively by the covalently linked β barrel domain or by a more complex channel that contains other components (e.g., the β barrel of BamA). We think that adding the words "through the UpaG barrel" would give readers the incorrect impression that the UpaG barrel is known to be the sole component of the secretion machinery.

2. Line 52: change "appear to partition" into "likely partition". I don't think many people would get upset about this.

We have made the suggested change.

3. Finally, regarding point 10, I maintain that the reference to unpublished data should be removed regardless of journal policy. How can "unpublished results provide useful information"?

As suggested we have removed the reference to unpublished data.